# Asprosin-neutralizing antibodies as a treatment for metabolic syndrome

Ila Mishra[1†], Clemens Duerrschmid[1†], Zhiqiang Ku[2], Yang He[3], Wei Xie[1], Elizabeth Sabath Silva[1], Jennifer Hoffman[1], Wei Xin[4], Ningyan Zhang[2], Yong Xu[3], Zhiqiang An[2], Atul R Chopra[1,5,6]*

[1]Harrington Discovery Institute, University Hospitals, Cleveland, United States; [2]Texas Therapeutics Institute, Brown Foundation Institute of Molecular Medicine, University of Texas Health Science Center at Houston, Houston, United States; [3]Baylor College of Medicine, Houston, United States; [4]Department of Pathology, Case Western Reserve University, Cleveland, United States; [5]Department of Medicine, Case Western Reserve University, Cleveland, United States; [6]Department of Genetics and Genome Sciences, Case Western Reserve University, Cleveland, United States

## Abstract

**Background:** Recently, we discovered a new glucogenic and centrally acting orexigenic hormone – asprosin. Asprosin is elevated in metabolic syndrome (MS) patients, and its genetic loss results in reduced appetite, leanness, and blood glucose burden, leading to protection from MS.

**Methods:** We generated three independent monoclonal antibodies (mAbs) that recognize unique asprosin epitopes and investigated their preclinical efficacy and tolerability in the treatment of MS.

**Results:** Anti-asprosin mAbs from three distinct species lowered appetite and body weight, and reduced blood glucose in a dose-dependent and epitope-agnostic fashion in three independent MS mouse models, with an IC50 of ~1.5 mg/kg. The mAbs displayed a half-life of over 3days in vivo, with equilibrium dissociation-constants in picomolar to low nanomolar range.

**Conclusions:** We demonstrate that anti-asprosin mAbs are dual-effect pharmacologic therapy that targets two key pillars of MS – over-nutrition and hyperglycemia. This evidence paves the way for further development towards an investigational new drug application and subsequent human trials for treatment of MS, a defining physical ailment of our time.

**Funding:** DK118290 and DK125403 (R01; National Institute of Diabetes and Digestive and Kidney Diseases), DK102529 (K08; National Institute of Diabetes and Digestive and Kidney Diseases), Caroline Wiess Law Scholarship (Baylor College of Medicine, Harrington Investigatorship Harrington Discovery Institute at University Hospitals, Cleveland); Chao Physician Scientist Award (Baylor College of Medicine); RP150551 and RP190561 (Cancer Prevention and Research Institute of Texas [CPRIT]).

*For correspondence: atul.chopra@case.edu

†These authors contributed equally to this work

## Introduction

Obesity and its co-morbidities, such as insulin resistance, hypertension, and dyslipidemia, are omnipresent, affecting nearly a quarter of the world population by some estimates (*Saklayen, 2018*). These conditions, which feed the spread of type II diabetes, coronary artery disease, stroke, nonalcoholic steatohepatitis, nephropathy, and other diseases, are commonly clustered under the umbrella term metabolic syndrome (MS) or syndrome X (*Saklayen, 2018*). MS is a consequence of chronic over-nutrition, turning the evolutionary drive to gather energy from the environment into a liability. As a whole, MS currently exists as an untreatable malady despite decades of basic research and drug development (*Saklayen, 2018*).

Through the study of a rare genetic condition in humans, neonatal progeroid syndrome (NPS, also known as marfanoid–progeroid–lipodystrophy syndrome), we recently discovered a fasting-induced, glucogenic, and orexigenic hormone that is the C-terminal cleavage product of profibrillin (encoded by FBN1) and named it asprosin (*Romere et al., 2016*). Its two major sites of action are the liver and the brain (*Romere et al., 2016*; *Li et al., 2019*; *Duerrschmid et al., 2017*). At the liver, asprosin causes a glucogenic effect through a cAMP-PKA-dependent pathway (*Romere et al., 2016*). It was found recently to promote hepatic glucose release through the binding of OR4M1, an olfactory G-coupled protein receptor in the rhodopsin family (*Li et al., 2019*). In addition, asprosin was shown to bind the mouse ortholog, Olfr734 with high affinity, and elimination of the receptor considerably reduced the glucogenic effects of exogenously administered asprosin (*Li et al., 2019*). There is also evidence, that asprosin crosses the blood brain barrier and exerts effects on the hypo-thalamus (*Duerrschmid et al., 2017*). In the arcuate nucleus of the hypothalamus, asprosin directly activates orexigenic AgRP neurons and indirectly inhibits anorexigenic POMC neurons, resulting in appetite stimulation. Patients with NPS, a human genetic model of deficiency in plasma asprosin, present with low appetite associated with extreme leanness and robust insulin sensitivity (*Romere et al., 2016*; *Duerrschmid et al., 2017*). NPS mutations in mice ($Fbn1^{NPS/+}$) result in phe-nocopy of the human disorder, and depressed AgRP neuron activity, which can be restored to nor-mal with asprosin replenishment in vivo and ex vivo (*Duerrschmid et al., 2017*). Importantly, $Fbn1^{NPS/+}$ mice are completely immune to diet-induced MS (*Duerrschmid et al., 2017*). On the opposite end of the energy-balance spectrum, patients and mice with MS exhibit elevated plasma asprosin (*Duerrschmid et al., 2017*; *Wang et al., 2020a*; *Ugur and Aydin, 2019*; *Alan et al., 2019*; *Zhang et al., 2019*; *Baykus et al., 2019*).

Based on these observations, we hypothesized that pharmacologic inhibition of asprosin is partic-ularly well suited to the treatment of MS, a condition in need of simultaneous reduction in both appetite and the blood glucose burden. Similar to humans, mice with MS display elevations in plasma asprosin (*Li et al., 2019*; *Duerrschmid et al., 2017*; *Baykus et al., 2019*), making them ideal preclinical models for testing this hypothesis.

To test this hypothesis, we generated three independent monoclonal antibodies (mAbs) that rec-ognize unique asprosin epitopes and investigated their preclinical efficacy and tolerability in the treatment of MS. We specifically dissected the suitability of acute and chronic asprosin neutralization in three different mouse models of MS – mice treated with a high-fat diet (diet-induced obesity [DIO]), mice with genetic leptin receptor mutation ($Lepr^{db/db}$), and mice treated with a diet rich in fructose and cholesterol in addition to fat (AMLN diet). For absolute proof-of-concept, we tested the ability of artificially induced plasma asprosin (adenovirus and adeno-associated virus mediated hepatic overexpression of a forcibly secreted form of asprosin) to raise blood glucose, appetite, and body weight, even in the presence of normal chow, followed by rescue of those parameters with mAb-mediated neutralization of asprosin. Our results demonstrate a promising treatment of MS with the use of anti-asprosin mAbs as a targeted, bimodal therapeutic strategy.

## Materials and methods

### Key resources table

| Reagent type (species) or resource | Designation | Source or reference | Identifiers | Additional information |
|---|---|---|---|---|
| Animal resource (*Mus musculus*) | C57BL/6J mice | Jackson Laboratory | RRID:IMSR_JAX: 000664 | |
| Animal resource (*Mus musculus*) | C57BL/6-Fbn1em1Chop/J | Jackson Laboratory | RRID:IMSR_JAX: 033548 | |
| Animal resource (*Mus musculus*) | C57BL/6J DIO | Jackson Laboratory | RRID:IMSR_JAX: 380050 | |
| Animal resource (*Mus musculus*) | B6.BKS(D)-Leprdb/J | Jackson Laboratory | RRID:IMSR_JAX: 000697 | |
| Research rodent food | Dustless pellet diet | Bio-Ser | F0173 | |

*Continued on next page*

*Continued*

| Reagent type (species) or resource | Designation | Source or reference | Identifiers | Additional information |
|---|---|---|---|---|
| Research rodent food | High-fat diet | Envigo Teklad | TD.06414 | |
| Research rodent food | AMLN diet | Research Diets | D09100301 | |
| Antibody | Mouse anti-asprosin mAb | This paper | | |
| Antibody | Rabbit anti-asprosin mAb | This paper | | |
| Antibody | Human anti-asprosin mAb | This paper | | |
| Antibody | Anti-mouse secondary antibody | Tonbo Biosciences | 72–8042 M001 | 1:10,000 |
| Antibody | Anti-rabbit secondary antibody | Cytiva's Amersham ECL | NA934-1ML | 1:10,000 |
| Adenoviral vector | Ad5-hFBN1 | This paper | | $3.6 \times 10^9$ pfu/mouse |
| Adenoviral vector | Ad5-hAsprosin | This paper | | $5 \times 10^{10}$ pfu/mouse |
| Adeno-associated viral vector | AAV8-hAsprosin | This paper | | $1 \times 10^{12}$ GC/mouse |
| Adenoviral vector | Ad5-Empty | This paper | | $3.6 \times 10^9$ pfu/mouse or $5 \times 10^{10}$ pfu/mouse |
| Adeno-associated viral vector | AAV8-Empty | This paper | | $1 \times 10^{12}$ GC/mouse |
| Commercial kit | cAMP ELISA kit | Crystal Chemicals | Catalog # 581001 | |
| Commercial kit | Glucose quantitation kit | Cayman Chemicals | Catalog # 10009582 | |
| Commercial kit | Mouse insulin ELISA kit | Crystal Chemicals | Catalog # 90080 | |
| Commercial kit | Human insulin ELISA kit | Raybiotech | Catalog # ELH-Insulin | |
| Commercial kit | IL1β ELISA kit | Abcam | Catalog # ab197742 | |
| Commercial kit | TNF ELISA kit | Abcam | Catalog # ab208348 | |
| Commercial kit | IL10 ELISA kit | Thermo Scientific | Catalog # BMS614INST | |
| Commercial kit | TGFβ ELISA kit | R and D systems | Catalog # DB100B | |
| Commercial kit | ALT ELISA kit | Abcam | Catalog # ab105134 | |
| Commercial kit | AST ELISA kit | Abcam | Catalog # ab263882 | |

## Mice

C57Bl/6J (wild-type, WT), *Fbn1*$^{NPS/+}$ mice (NPS), DIO, and leptin-receptor-deficient (*Lepr*$^{db/db}$) mice were purchased from Jackson Laboratories. Mice were fed normal chow (5V5R, Lab Supply), dustless

pellet diet (F0173, Bio-Serv), high-fat diet (60% calories from fat, TD.06414, Envigo Teklad), or the Amylin diet rich in fructose and cholesterol in addition to fat (AMLN diet, D09100301, Research Diets), where indicated. Twelve-week-old C57Bl/6J WT male mice were used for adenovirus and adeno-associated virus studies and lean mice anti-asprosin mAb studies. Sixteen-week-old mice with DIO, maintained on high-fat diet for 12 weeks, were used for single-dose anti-asprosin mAb experiments, assessment of glucosuria, hepatic cAMP levels, mAb treatment-associated side effects, mAb half-life, and plasma markers of lipodystrophy assessment. Twenty-five- to thirty-week-old DIO mice were used for 14 days mAb study and MRI assessment. Thirty-week-old C57Bl/6J WT mice, maintained on 24 weeks of AMLN diet, and young (12-week-old) and old (30-week-old) $Lepr^{db/db}$ mice were used for chronic mAb experiments. Thirty-five-week-old $Fbn1^{NPS/+}$ mice were used for the acute mAb experiment. All experimental mice were age and sex matched with control mice across all experiments in this study.

## mAb generation

A majority of the studies were conducted with a mouse mAb (M1). M1 was generated using traditional hybridoma techniques by immunizing mice with a 28 amino acid peptide KKKELNQLED RYDKDYLSGELGDNLKMK located close to the C-terminus of asprosin. Where indicated only, rabbit and fully human anti-asprosin mAbs were used in parallel with the mouse anti-asprosin mAb. The rabbit mAb was generated by immunizing rabbits with recombinant full-length human asprosin at RevMAb Biosciences, USA, and cloning variable region genes from positive single memory B cells based on protocols described previously (*Gui et al., 2019*). A fully human mAb was generated from a naïve human phage display antibody library by panning against recombinant full-length human asprosin (Texas Therapeutics Institute at the University of Texas Health Science Center at Houston). All three mAbs display cross-reactivity to mouse and human asprosin.

## mAb injection

Mice received either a single dose of 250 µg/mouse (~5–6 mg/kg) mAb intra-peritoneally in 500 µl USP-grade saline, or repeated daily dose for up to 21 days. Injections were performed between 9 am and 11 am both for single- and repeated-dose studies. For dose–response curve, DIO mice received a single IP injection of different doses of control, isotype-matched IgG, or anti-asprosin mouse mAb (15.63, 31.25, 62.5, 125, and 250 µg/mouse, corresponding to 0.4, 0.85, 1.64, 3.28, and 6.88 mg/kg; n = 5/dose) at 11–11:30 am.

## Adenovirus and adeno-associated virus experiments

Twelve-week-old C57Bl/6J mice were injected intravenously via the tail-vein with adenovirus (Ad5) or adeno-associated virus, serotype 8 (AAV8) dissolved in 150 µl USP-grade sterile saline. Mice injected with Ad5-empty ($3.6 \times 10^9$ pfu/mouse) served as controls for experimental mice that received Ad5-FBN1 virus ($3.6 \times 10^9$ pfu/mouse) containing the human FBN1 coding region under control of a CMV promoter. Mice injected with Ad5-empty ($5 \times 10^{10}$ pfu/mouse) served as controls for experimental mice that received Ad5-IL2-Asprosin ($5 \times 10^{10}$ pfu/mouse) containing an N-terminal his-tagged human asprosin coding region preceded by an IL2 signal peptide, under control of an EF1 promoter. Mice injected with AAV8-empty ($1 \times 10^{12}$ GC/mouse) served as controls for experimental mice that received AAV8-IL2-asprosin ($1 \times 10^{12}$ GC/mouse) containing an N-terminal his-tagged human asprosin coding region preceded by an IL2 signal peptide, under control of an EF1 promoter.

## Food intake

Mice were placed in CLAMS metabolic cages (Columbus Instruments) and acclimated for 3 days. Food intake was recorded for 24 hr, and mice were returned to normal housing. Food intake was manually measured in 24 hr experiments in DIO mice and Ad5 and AAV8 studies. In Ad5 and AAV8 studies, mice were singly housed in standard caging and fed a pelleted, dustless diet (F0173, Bio-Serv). In single day DIO experiments, mice were acclimated to crushed high-fat diet (60% calories from fat, TD.06414, Envigo Teklad) in single housing. The diet was replenished, weighed, and re-weighed after 24 hr to establish food intake.

## Magnetic resonance imaging

Twenty-five-week-old DIO mice were acclimated to the Case Center for Imaging Research Facility, Cleveland, OH, for a day prior to each imaging session. Mice were subjecting to MRI scan on day 0 and day 14 of control IgG and anti-asprosin mAb treatment (250 µg in 500 µl 0.9% saline). Immediately prior to imaging, mice (n = 7 and 8 for each group) were weighed and anesthetized with 2–3% isoflurane in oxygen. Lean mass, fat mass, and overall body weight were calculated as previously described (*Bederman et al., 2018*). Briefly, the anesthetized mice were placed in a prone position within a Bruker Biospec 7T MRI scanner (Bruker Biospin, Billerica, MA). A 72-mm-diameter volume coil was used for excitation and signal detection to maximize the uniformity of the images. After localizer scans, a relaxation compensated fat fraction (RCFF) MRI acquisition and reconstruction process was used to generate quantitative fat fraction maps for each imaging slice. A semiautomatic image analysis was performed to generate separate fat and water images for each imaging slice. Total fat volume and lean body volume were calculated from a compilation of the fat fraction images and each animal's weight. The fat fraction and water fraction images were then segmented to calculate the respective volumes of peritoneal and subcutaneous adipose tissues.

## Plasma parameters

Mouse glucose was determined using a hand-held glucometer (OneTouch Ultra2, LifeScan) from a droplet of tail blood, and insulin was measured using Crystal Chem (catalog # 90080) mouse insulin ELISA kit. Mouse plasma total cholesterol, HDL (high-density lipoprotein cholesterol), LDL (low-density lipoprotein cholesterol), triglyceride (TG), free fatty acids (FFA), and glycerol levels were measured by the mouse metabolism and phenotyping core at Baylor College of Medicine, Houston, TX (MMPC at BCM and NIH fund RO1DK114356 and UM1HG006348). Hepatic cAMP levels were determined using CrystalChem (catalog # 581001) cAMP Elisa kit. Glucose and insulin levels in human plasma purchased from BioIVT were measured using a chromogenic glucose quantitation kit (Cayman Chemicals; catalog # 10009582) and a human insulin ELISA kit (Raybiotech; catalog # ELH-Insulin), respectively. Pro-inflammatory (IL1β [Abcam, catalog # ab197742] and TNFα [Abcam, catalog # ab208348]) and anti-inflammatory (IL10 [Thermo Scientific, catalog # BMS614INST] and TGFβ [R and D systems, catalog #DB100B]) cytokines were measured in plasma collected from 16-week-old male DIO mice intra-peritoneally injection of control, isotype-matched IgG, or anti-asprosin mouse mAb (250 µg mAb in 500 µl 0.9% saline). Urine glucose (glucosuria, CrystalChem, catalog # 81692), plasma alanine transaminase activity (ALT, Abcam, catalog # ab105134), and aspartate amino aminotransferase (AST, Abcam, catalog # ab263882), plasma creatine and blood urea nitrogen (BUN) levels were measured in plasma collected after intra-peritoneal injection of isotype-matched IgG or anti-asprosin mouse mAb (250 µg mAb in 500 µl 0.9% saline) in 16-week-old male DIO mice. Plasma creatine and BUN were measured by the mouse metabolism and phenotyping core at Baylor College of Medicine, Houston, TX (MMPC at BCM and NIH fund RO1DK114356 and UM1HG006348).

All samples were run in duplicates across assays. To gauge the overall reliability of ELISA results, threshold for acceptable variability was set at 15% for inter- and intra-assay CV (coefficient of variance).

## Locomotor activity

Sixteen-week-old male DIO mice were acclimated to single housing in Promethion metabolic cages (Sable Systems International) for 24 hr. Thereafter, two parameters of locomotor activity, pedestrian activity and wheel running activity, were recorded, 9 hr prior and 36 hr post-intra-peritoneal injection of control, isotype-matched IgG, or anti-asprosin mAb (250 µg mAb in 500 µl 0.9% saline).

## Half-life of asprosin-neutralizing mAbs

For assessing in vivo half-life of mAbs, mouse models of 'high asprosin', C57Bl/6J (WT) mice with DIO (n = 6), and 'low asprosin' (NPS mice, n = 6) were injected with mouse mAb (n = 2/group), rabbit mAb (n = 2/group), or human mAb (n = 2/group) (250 µg mAb in 500 µl 0.9% saline). Thereafter, mAb levels were determined in mouse plasma collected at 2, 6, 24, 96, 336, and 504 hr post-injection.

## Asprosin and anti-asprosin mAb ELISA detection procedures

For assessing in vivo half-life of antibodies, antibody in plasma was captured on ELISA plate coated with asprosin (100 ng/well), and levels of mouse mAb, rabbit mAb, and human mAb were detected with HRP-conjugated anti-mouse (1:10,000), anti-rabbit (1:10,000), or anti-human (1:10,000) secondary antibodies, respectively.

For detection of endogenous asprosin in human plasma samples and adeno and adeno-associated virus generated human asprosin in mouse plasma, a human asprosin sandwich ELISA was custom built using mouse monoclonal anti-asprosin antibody against human asprosin amino acids 106–134 (human profibrillin amino acids 2838–2865) as the capture antibody and a rabbit anti-asprosin monoclonal antibody as the detection antibody. An anti-rabbit secondary antibody linked to HRP was used to generate a signal, and mammalian-cell produced recombinant human asprosin was used to generate a standard curve. Human asprosin was detected in 10 µl of EDTA-treated human plasma. For detection of human asprosin in plasma of mice treated with human asprosin expressing adeno and adeno-associated viruses, plasma samples were first processed for IgG and albumin removal using Proteome purify2 columns (R and D Systems, catalog # IDR002) and concentrated using vivaspin 500 (VS0131) PES filters before running the ELISA.

Additional characterization of rabbit and human anti-asprosin mAb was done as previously described for the mouse mAb (*Duerrschmid et al., 2017*). Briefly, recombinant asprosin (1 nM) was preincubated with increasing concentrations of anti-asprosin rabbit and human mAb (0, 0.007, 0.3, and 0.215 µg/nM asprosin) for 1 hr at room temperature. As a control, recombinant asprosin (1 nM) was preincubated with increasing concentrations of isotype-matched IgG antibody. Thereafter, free asprosin was detected using sandwich ELISA as follows: free asprosin in the mixture of recombinant asprosin preincubated with increasing dose of the rabbit mAb or IgG was captured with the rabbit mAb and detected with the anti-asprosin mouse mAb. HRP-labeled anti-mouse secondary antibody was used to generate a signal. Similarly, free asprosin in the mixture of recombinant asprosin preincubated with increasing dose of the human mAb or IgG was captured with the human mAb and detected with the anti-asprosin rabbit mAb. HRP-labeled anti-rabbit secondary antibody was used to generate a signal.

## Epitope competition

Epitope competition assay was done to determine whether the three mAbs recognized the same epitope or different epitopes. For this, we captured recombinant asprosin (20 nM) on an ELISA plate coated with one of the three mAbs (100 ng/well), followed by a detection with each of the three mAbs (100 ng/well, 3 × 3 matrix), mouse, rabbit, or human mAb. Subsequently, captured asprosin was then incubated in parallel with each of the three different mAbs (100 ng/well, 3 × 3 matrix). Depending on which species antibody was used in the second step, a species-matched secondary detection antibody was used. For example, each well that was incubated with a mouse anti-asprosin mAb in the second step was then incubated with an anti-mouse secondary HRP-conjugated antibody. Mouse/mouse, rabbit/rabbit, and human/human pairs served as internal positive control. For clarity, only one version of each combination is presented here, but the competition worked both ways.

## Asprosin-mAb binding affinity measurement with BLI (Bio-layer Interferometry)

Antibody affinity was measured on Pall ForteBio's Octet RED96 system. Recombinant human asprosin (25 mg/ml) was loaded onto a Ni-NTA biosensor (ForteBio, catalog # 18–5102) for 300 s. Following 20 s of baseline in kinetics buffer (ForteBio, catalog # 18–5032), the loaded biosensor was dipped in parallel into a series of antibody solutions on increasing concentrations (0.14–300 nM) for 300 s to record the association kinetics and then dipped into the kinetic buffer (ForteBio, catalog # 18–5032) for 600 s to record the dissociation kinetics. Kinetic buffer without antibody was set to correct the background. $K_D$ was determined from seven kinetic curves fitted in a 1:1 binding model using global fitting in ForteBio's data analysis software.

## AgRP+ neuron labeling and electrophysiology

To identify AgRP+ neurons, we crossed the Rosa26–tdTOMATO mice with Agrp–Cre mice to generate Agrp–Cre ×Rosa26–tdTOMATO mice, which express TOMATO selectively in AgRP+NPY+ neurons (*Duerrschmid et al., 2017*). AgRP neuron identification, electrophysiological recording in response to recombinant asprosin, and recombinant asprosin preincubated with anti-asprosin monoclonal Abs was done as previously described (*Duerrschmid et al., 2017*). Briefly, mice were deeply anesthetized with isoflurane and were transcardially perfused with a modified ice-cold sucrose based cutting solution (pH 7.3) (*Duerrschmid et al., 2017*). Slices (250 µm) were cut with a Microm HM 650V vibratome (Thermo Scientific). Three brain slices containing the ARH were obtained for each mouse and recordings were made at levels throughout this brain region. The slices were recovered for 1 hr at 34°C and then maintained at room temperature in artificial cerebrospinal fluid (aCSF, pH 7.3) that was saturated with 95% $O_2$ and 5% $CO_2$ before recording. TOMATO-labeled neurons in the ARH were visualized using epifluorescence and infrared–differential interference contrast (IR-DIC) imaging on an upright microscope (Eclipse FN-1, Nikon). Recordings were made using a Multi-Clamp 700B amplifier (Axon Instruments), sampled using Digidata 1440A, and analyzed offline with pClamp 10.3 software (Axon Instruments). Series resistance was monitored during the recording, and the values were generally 2 mV increase of the resting membrane potential, whereas hyperpolarization was defined as a >2 mV decrease of the resting membrane potential; values between a 2 mV decrease and a 2 mV increase was defined as 'irresponsive'. rAsprosin and either anti-asprosin or IgG were gently mixed at a 1:100 ratio and then kept on ice for 1 hr before recording. After AgRP + neuron response to asprosin alone was confirmed, the mixture of rAsprosin and the asprosin-specific mAb or IgG was perfused to treat the AgRP+ neurons for 4 min.

## Statistical analysis

Data was graphed and analyzed using GraphPad Prism (Version six and higher). Data is presented as mean ± standard error of the mean. Depending on the format of data, analysis was by using t-test (two groups, one time point), or analysis of variance (ANOVA) sets involving multiple groups and time points. ANOVA is indicated with the word 'ANOVA' in figures next to the significance asterisk. Non-linear three and four parametric variable slope (least square regression) analyses determined the $IC_{50}$ and R (*Romere et al., 2016*) values in dose–response studies. For statistical comparisons, data points exclusion criteria were set at Q = 1% for grubbs' outlier analysis and a CV% of >15% between sample duplicates run on an ELISA. Original source data, grubbs' outlier analysis values, and details of statistical analysis are presented as supplementary source data file. Significance is presented using the asterisk symbol (*$p<0.05$, **$p<0.01$, ***$p<0.001$, and ****$p<0.0001$). In situations where a non-significant trend was observed, the full p-value is presented.

# Results

## A single dose of an asprosin neutralizing mAb reduces appetite, body weight, and blood glucose levels in mice with MS

We previously showed that adolescent boys with MS have elevated circulating asprosin and that NPS patients have undetectable plasma asprosin (*Romere et al., 2016*; *Duerrschmid et al., 2017*). Here we find that a cohort of adult men with MS have significantly elevated plasma asprosin, with levels more than twofold higher than those of unaffected age- and sex-matched individuals (*Figure 1—figure supplement 1d*). This result is consistent a multitude of independent studies (*Duerrschmid et al., 2017*; *Wang et al., 2020a*; *Ugur and Aydin, 2019*; *Alan et al., 2019*; *Zhang et al., 2019*; *Baykus et al., 2019*).

To generate a preclinical model of MS, we exposed C57BL/6 mice to a high-fat diet for a minimum of 12 weeks (DIO mice). A single dose of an asprosin-neutralizing mAb was sufficient to significantly reduce food intake by an average of 1 g/day in the treatment group, but not in mice receiving a control, isotype-matched IgG (*Figure 1a*). This decrease in food intake was associated with a 0.7 g average reduction in body weight over 24 hr (*Figure 1b*). In mice without access to food, a significant blood glucose reduction was evident as early as 2 hr post-mAb treatment. The effect was most pronounced at 4 hr and 6 hr after injection (*Figure 1c*). Importantly, decreased blood glucose was not paralleled by increased glucosuria in response to mAb treatment (*Figure 1—figure supplement*

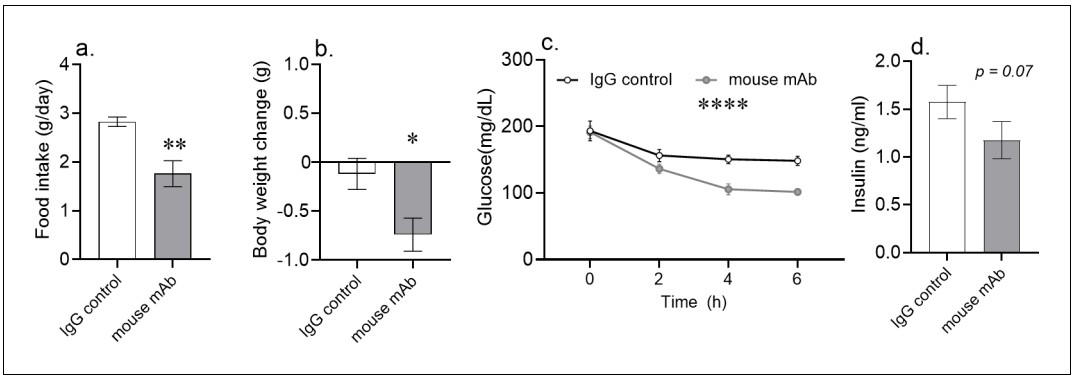

**Figure 1.** Acute asprosin neutralization reduces plasma glucose, appetite, and body weight in diet-induced obese mice. (**a–d**) Cumulative food intake and body weight change (measured 24 hr post-treatment), baseline blood glucose (measured at hour 2, 4, and 6 post-mAb treatment) and plasma insulin (measured 6 hr post-treatment) were measured after a single dose of anti-asprosin mAb (250 µg/mouse) in 16-week-old male DIO (diet-induced obesity) mice, n = 5 or 6/group. Note that mice were without food for the duration of the experiment in (**c, d**), demonstrating that the glucose lowering effect was independent of the hypophagic effect of mAb treatment. Asterisk (*) indicate the range of alpha as determined by the t-test (two groups, one time point), or analysis of variance (ANOVA, sets involving multiple groups and time points. *p<0.05, **p<0.01, ***p<0.001, and ****p<0.0001; *Figure 1—source data 1*).

The online version of this article includes the following source data and figure supplement(s) for figure 1:

**Source data 1.** Raw data and statistical analysis values for *Figure 1* and *Figure 1—figure supplement 1*.

**Figure supplement 1.** Higher blood glucose and insulin levels in metabolic syndrome patients are associated with elevated asprosin levels.

**Figure supplement 2.** Asprosin neutralization does not cause glucosuria.

**Figure supplement 3.** Acute asprosin neutralization reduces blood glucose, but not appetite and body weight in lean mice.

**Figure supplement 4.** Blood glucose, appetite, and body weight remain unaltered in response to anti-asprosin treatment in Fbn1[NPS/+] mice.

**Figure supplement 5.** Anti-asprosin monoclonal antibody treatment does not lead to confounding side effects.

*2*), ruling glucosuria out as a potential mechanism for blood glucose reduction. Also, given that mice were without access to food, this result demonstrates that the blood glucose lowering effect is independent of reductions in food intake.

Interestingly, repeating this experiment in lean C57BL/6 mice (with normal plasma asprosin compared with DIO mice) showed a subtle effect on blood glucose without inducing hypoglycemia, and no effect at all on 24 hr cumulative food intake or body weight (*Figure 1—figure supplement 3*), suggesting that elevation in plasma asprosin or presence of florid MS may be necessary prerequisites to unlocking the full therapeutic effect of asprosin neutralization. This result also argues against potential toxicity or off-target effects of the mAb being responsible for the anti-MS effects observed in other models, since any such toxicity or off-target effects should have also manifested in lean mice. Furthermore, NPS (*Fbn1[NPS/+]*) mice, with genetic loss of asprosin, were completely unresponsive to mAb treatment, with no effect at all on blood glucose, 24 hr cumulative food intake, and body weight (*Figure 1—figure supplement 4*). These results again suggest that the glucose and appetite reducing actions of this particular anti-asprosin mAb are specifically due to asprosin neutralization, and not potential off-target or toxic effects.

To further probe potential toxicity or off-target effects of asprosin neutralization, we tested if anti-asprosin mAb treatment has any confounding side effects. Markers of general sickness (pedestrian locomotor and intentional wheel running activity), plasma levels of pro-inflammatory (IL1β and TNFα) and anti-inflammatory (IL10 and TGFβ) cytokines, markers of renal (plasma creatinine and BUN), and hepatic health (plasma ALT, alanine aminotransferase and AST, aspartate aminotransferase) were unaltered in DIO mice treated with the anti-asprosin mAb (*Figure 1—figure supplement 5*), suggesting lack of a broad toxicity signature.

## Acute asprosin neutralization dose dependently mitigates MS

A single dose of the mAb resulted in a significant reduction in blood glucose levels (measured 4 hr post-treatment), in plasma insulin and triglyceride levels (measured 24 hr post-treatment), and in 24 hr cumulative food intake and body weight in a dose-dependent manner in DIO mice (*Figure 2a–e*). Half maximal inhibitor concentration (IC$_{50}$) determined using four-parameter non-linear variable slope curve for these three measures was in the range of 30–55 μg/mouse (~1.5 mg/kg). An isotype-matched, control IgG did not show any effect at all. Interestingly, the dose–response curve shapes were distinct for the five endpoints, suggesting that either the potency of asprosin varies among these measures or they utilize distinct epitopes on asprosin that are differentially affected by this mAb. Furthermore, the plateauing of the mAb effect for body weight, glucose, insulin, and triglycerides with higher doses suggests the existence of internal compensatory/buffer mechanisms to protect against drastic reductions in those parameters.

## Viral overexpression of human asprosin induced a hyperphagic, obese, and hyperglycemic phenotype in mice that was rescued by mAb-mediated neutralization

We were interested in the effects of sustained elevation of asprosin in mice fed a normal diet, and in generating tools that obviate the need to employ recombinant proteins of variable activity. To this end, we compared three gain-of-function experiments where we transduced mice with adenoviruses (Ad5) or adeno-associated viruses (AAV8) encoding human *Fbn1* (with native signal peptide) or human cleaved asprosin (with an IL2 signal peptide to promote secretion), and then tested whether viral vector-induced metabolic phenotypes could be rescued with mAb treatment. When compared to mice transduced with Ad5-empty and AAV8-empty control viral vectors, mice transduced with Ad5-*FBN1*, Ad5-Asprosin, and AAV8-Asprosin all exhibited hyperphagia, gained weight, and displayed significantly higher baseline glucose and insulin levels (*Figure 3a,e,i*; *Figure 3—figure supplement 1a–d,f–i,k–n*; *Figure 3—figure supplement 2a–c*). This metabolic phenotype in all three experiments was coincident with the elevation of human asprosin in mouse plasma in the time it takes to achieve peak expression in vivo (*Figure 3—figure supplement 1e,j,o*; *Crystal, 2014*; *Zincarelli et al., 2008*). For example, AAV8 vectors typically take up to 40–60 days to achieve peak

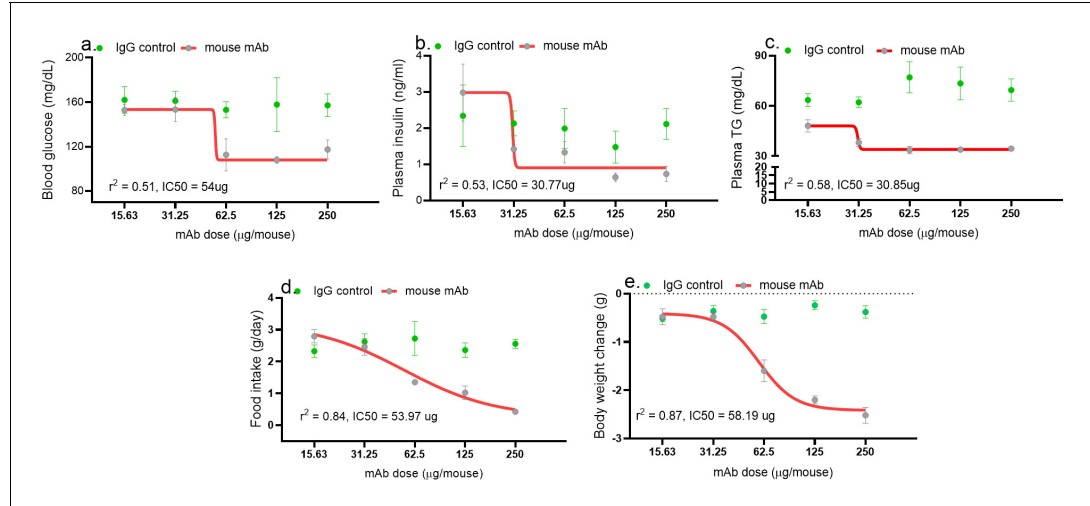

**Figure 2.** Asprosin neutralization corrects hyperglycemia, hyperphagia, and hypertriglyceridemia in a dose-dependent manner. Baseline blood glucose (measured 4 hr post-treatment), plasma insulin and triglyceride (TG; at 24 hr post-treatment) levels, cumulative food intake, body weight change (measured 24 hr post-treatment), were measured upon injecting increasing dose of anti-asprosin mAb in 16-week-old male DIO (diet-induced obesity) mice, n = 5/group. The doses tested were 15.63, 31.25, 62.5, 125, and 250 μg/mouse, corresponding to 0.4, 0.85, 1.64, 3.28, and 6.88 mg/kg. Half maximal inhibitor concentration (IC$_{50}$) was determined using a four- and three-parameter non-linear variable slope curve. r (*Romere et al., 2016*) value determined the goodness of fit (*Figure 2—source data 1*).

The online version of this article includes the following source data for figure 2:

**Source data 1.** Raw data and statistical analysis values for *Figure 2* and Figure 2-supplements.

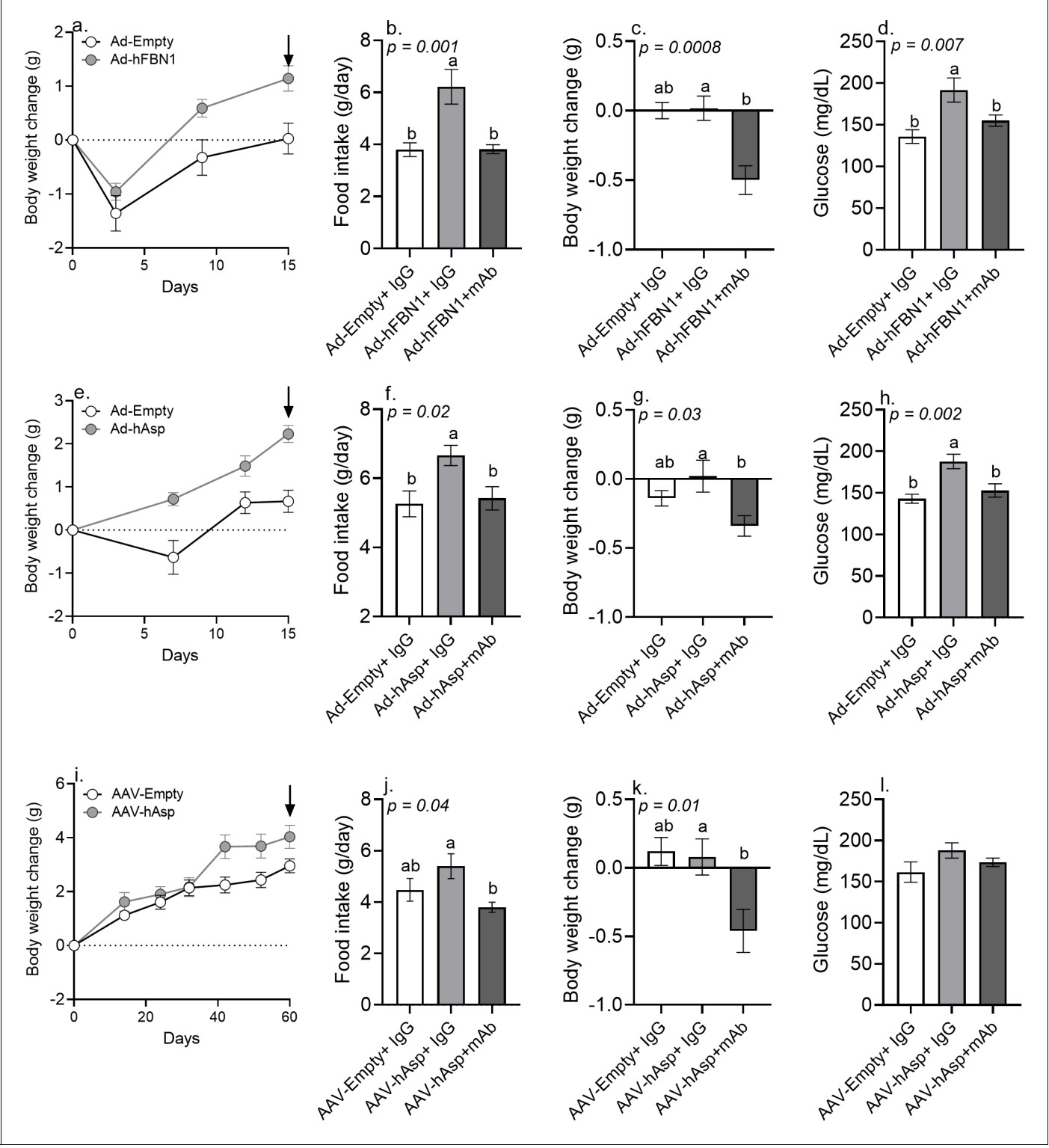

**Figure 3.** Immunologic neutralization fully rescues the metabolic effects of viral induction of plasma asprosin. (**a**) Body weight change was measured over 15 days after 12-week-old male C57Bl/6 mice were tail-vein-injected with Ad-empty or Ad-*FBN1* ($3.6 \times 10^9$ pfu/mouse, n = 12/group) viruses. Downward arrow indicates the day of mAb treatment described below. (**b–d**) Cumulative food intake, body weight change, and blood glucose were measured 24 hr after intra-peritoneal injection of indicated control, isotype-matched IgG, or anti-asprosin mAbs (n = 6/group) in the above mice. (**e**) Body weight change was measured over 15 days after 12-week-old male C57Bl/6 mice were tail-vein-injected with Ad-empty or Ad-asprosin ($5 \times 10^{10}$ pfu/mouse, n = 12/group) viruses. Downward arrow indicates the day of mAb treatment described below. (**f–h**) Cumulative food intake, body weight

*Figure 3 continued on next page*

*Figure 3 continued*

change, and blood glucose were measured 24 hr after intra-peritoneal injection of indicated control, isotype-matched IgG or anti-asprosin mAb (n = 6/group) in the above mice. (i) Body weight change was measured over 60 days after 12-week-old male C57Bl/6 mice were tail-vein-injected with AAV8-empty or AAV8-asprosin (1 × 10$^{12}$ GC/mouse, n = 10/group) viruses. Downward arrow indicates the day of mAb treatment described below. (j–l) Cumulative food intake, body weight change, and blood glucose were measured 24 hr after intra-peritoneal injection of indicated control, isotype-matched IgG, or anti-asprosin mAbs (n = 5/group) in the above mice. Different and same alphabets on bars indicate presence or absence of significant difference, respectively, between groups, as determined by one-way ANOVA followed by Dunnett's multiple comparisons test. p<0.05 was considered statistically significant (*Figure 3—source data 1*).

The online version of this article includes the following source data and figure supplement(s) for figure 3:

**Source data 1.** Raw data and statistical analysis values for *Figure 3* and figure 3-supplements.
**Figure supplement 1.** Viral overexpression of human asprosin results in a MS-like phenotype in lean mice.
**Figure supplement 2.** Raw body weight values of Ad5 and AAV experiments.

protein expression in vivo (*Zincarelli et al., 2008*). Notably, that is when the mice transduced with AAV8-*Asprosin* first display an increase in body weight (*Figure 3i*, *Figure 3—figure supplement 2c*). In contrast, Ad5 vectors produce a more accelerated peak expression, resulting in quicker increase in food intake and body weight (*Figure 3a,e*; *Figure 3—figure supplement 2ca, b*). In either case, these three distinct gain-of-function experiments clearly showed a significant increase in circulating asprosin (*Figure 3—figure supplement 1e,o*) and the associated phenotypes in an experimentally robust and rigorous manner.

When injected with the anti-asprosin mAb, mice with viral overexpression of human asprosin restored their food intake back to normal (*Figure 3b,f,j*), and this resulted in an average net body weight loss of 0.3–0.5 g within the first 24 hr (*Figure 3c,g,k*). A single injection of mouse anti-asprosin mAb also reduced baseline glucose levels back to normal (*Figure 3d,h,i*). The results of this combined gain-of-function/rescue study indicate that viral overexpression of asprosin is a valuable tool to study asprosin effects in non-obese mice and that mAb-mediated neutralization fully rescues the metabolic effects of elevated plasma asprosin.

## Chronic asprosin neutralization mitigates hyperphagia, hyperglycemia, and weight gain in obese mice

In DIO mice treated with daily injection for 10 days, we observed a ~10% decrease in body weight with asprosin neutralization (*Figure 4a*) that was associated with improved glucose tolerance on day 11 (after 1 day without mAb) (*Figure 4b*). On day 13 however (10 days of once daily mAb treatment followed by 3 days without mAb treatment), the body weight difference between the two groups remained the same (*Figure 4c*), but the improved glucose tolerance disappeared (*Figure 4d*), showing once again that the effect of asprosin neutralization on glucose homeostasis is independent of systemic improvements in metabolism due to weight loss, and also demonstrating the short effect-life of the mAb (24–48 hr).

We followed up the single-dose and 10 day studies by testing the effect of asprosin neutralization for 2 weeks. A 14 day course of daily injection of an anti-asprosin mAb significantly reduced food intake and body weight in diet-induced MS (DIO mice; *Figure 4e,f*; *Figure 4—figure supplement 1a*; *Figure 4—figure supplement 2a,c,d*), genetic MS (*Lepr*$^{db/db}$ mice; *Figure 4i,j*; *Figure 4—figure supplement 1b*), as well as in mice on a diet rich in fructose and cholesterol in addition to fat (AMLN diet) (*Figure 4m,n*; *Figure 4—figure supplement 1c*). On average, mice treated with the mAb weighed approximately 5–11% less than mice treated with a control mAb, indicating that the observed reduced food intake results in a net calorie deficit and weight loss. The reduction in body weight and food intake in response to 14 day mAb treatment in DIO mice was coincident with a reduction in subcutaneous and visceral fat depots, with no effect observed on lean mass (*Figure 5*).

Furthermore, while all three mouse models of MS displayed improvement in hyperglycemia (*Figure 4g,k,o*), a significant reduction in plasma insulin after mAb treatment was observed only in DIO mice (*Figure 4h,l,p*). Interestingly, while asprosin neutralization was unequivocally protective against obesity irrespective of the model of MS, the direction and magnitude of response varied. Thirty-week-old *Lepr*$^{db/db}$ mice, mice that had reached the plateaued phase of weight gain (*Wang et al., 2020b*), showed significant weight loss in response to the treatment (*Figure 4—figure supplement 2a,b*). On the other hand, 12-week-old *Lepr*$^{db/db}$ mice that are still in the linear phase

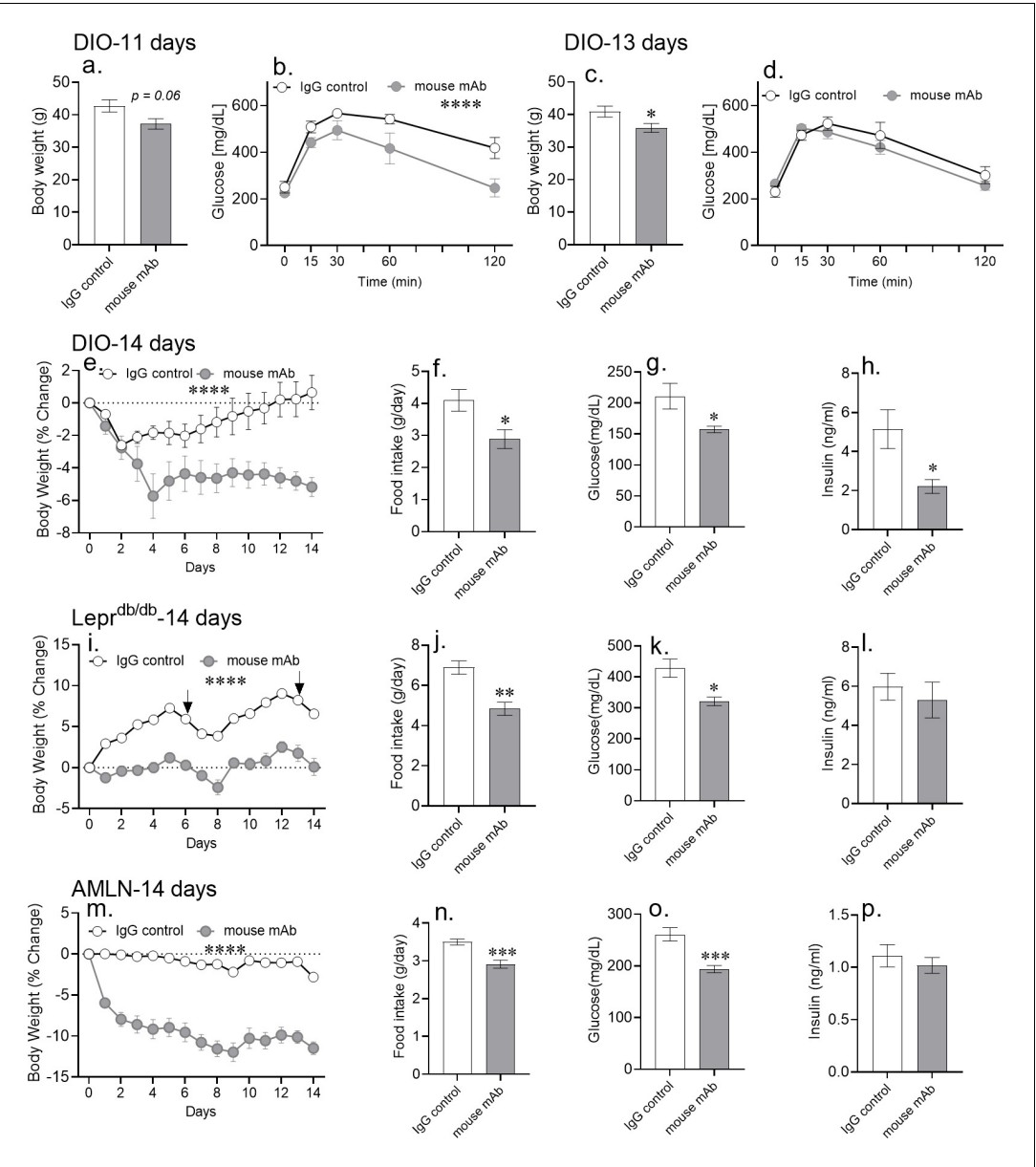

**Figure 4.** Chronic asprosin neutralization improves metabolic syndrome in three independent mouse models. (a–d) Body weight and glucose tolerance were measured on day 11 (**a, b**) and day 13 (**c, d**) after 10 days of once daily intra-peritoneal injection of control, isotype-matched IgG, or anti-asprosin mAb in 16-week-old male DIO (diet-induced obesity) mice (n = 5/group). (**e–p**) Percent change in body weight, 24 hr cumulative food intake (measured on day 7), and blood glucose and plasma insulin (6 hr post-treatment on day 14) levels were measured after 14 days of once daily intra-peritoneal injection of control, isotype-matched IgG, or anti-asprosin mAb in 30-week-old male DIO mice (n = 8 or 9/group; **e–h**), 12-week-old male *Lepr*<sup>db/db</sup> mice (n = 5 or 6/group; **i–l**), and 30-week-old male mice on AMLN diet (n = 7/group; **m–p**). Asterisk (*) indicate the range of alpha for an effect of mAb treatment as determined by the t-test (two groups, one time point), or two-way analysis of variance (two-way ANOVA, sets involving multiple groups and time points. *p<0.05, **p<0.01, ***p<0.001, and ****p<0.0001; *Figure 4—source data 1*). Downward arrows in (**m**) indicate the day of submandibular bleed in *Lepr*<sup>db/db</sup> mice. The online version of this article includes the following source data and figure supplement(s) for figure 4:

**Source data 1.** Raw data and statistical analysis values for figure and figure-supplements.
**Figure supplement 1.** Raw body weight values of 14 day mAb experiments.
**Figure supplement 2.** Chronic asprosin neutralization reduces body weight in LepR<sup>db/db</sup> and DIO mice.
**Figure supplement 3.** Asprosin neutralization improves dyslipidemia in MS mouse models.

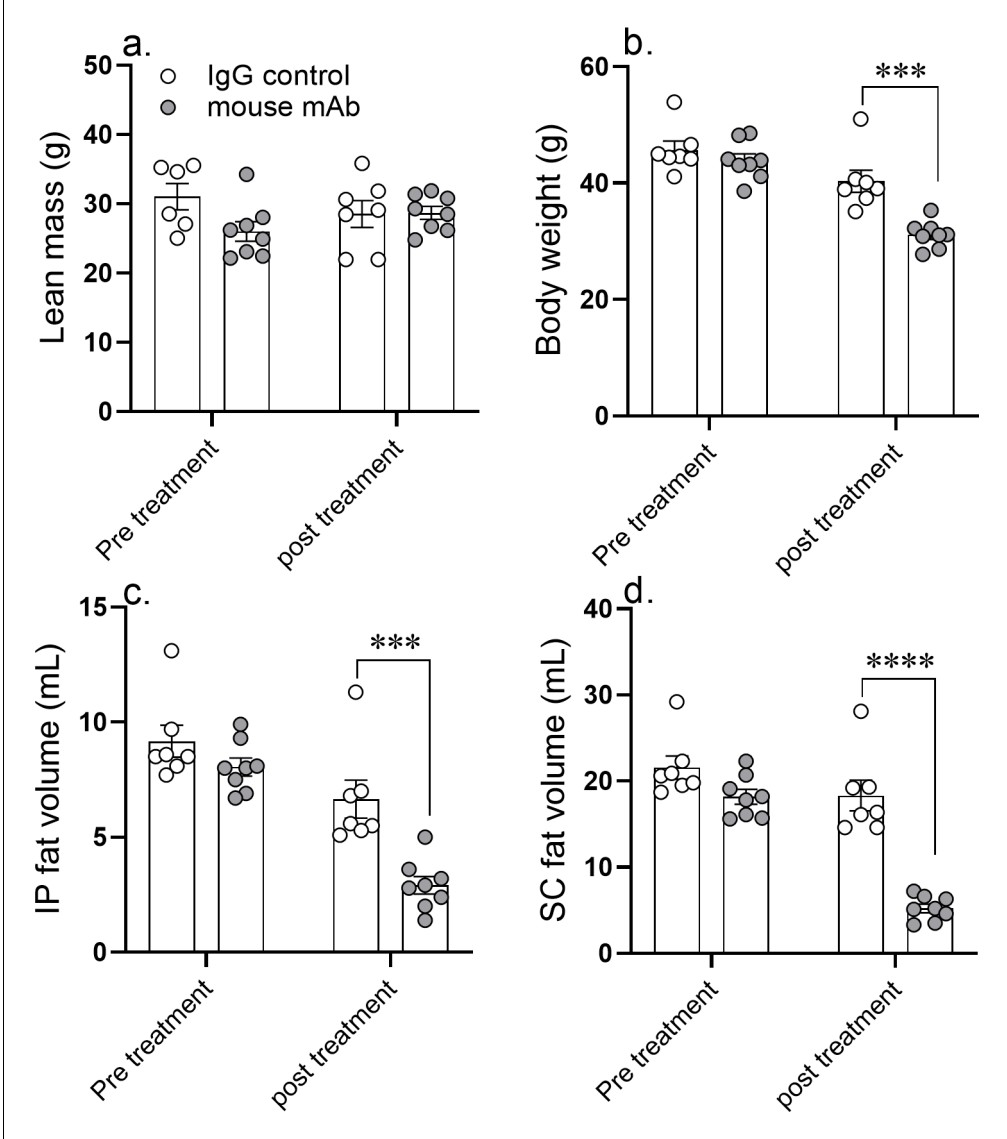

**Figure 5.** A 14 day course of anti-asprosin mAb treatment reduces adipose mass in diet-induced obese mice. Lean mass (**a**), body weight (**b**), intra peritoneal fat volume (**c**), and subcutaneous fat volume (**d**) were measured pretreatment (day 0) and post-treatment (day 14) of once daily intra-peritoneal injection of control, isotype-matched IgG, or anti-asprosin mAb in 25-week-old male DIO (diet-induced obesity) mice (n = 7 or 8/group). Asterisk (*) indicate the range of alpha as determined by the two-tailed Student's t-test (**Figure 5—source data 1**). *p<0.05, **p<0.01, ***p<0.001, and ****p<0.0001.

The online version of this article includes the following source data for figure 5:

**Source data 1.** Raw data and statistical analysis values for **Figure 5** and figure 5-supplements.

of weight gain (**Wang et al., 2020b**), showed prevention of weight gain rather than frank weight loss during the treatment period (**Figure 4i**).

A single injection of the anti-asprosin mAb significantly reduced plasma levels of total cholesterol, LDL, glycerol, and triglycerides in DIO mice, measured 6 hr after injection in mice without access to food (**Figure 4—figure supplement 3a–f**). Chronic treatment for 2 weeks however, in DIO, *Lepr*[db/db], and AMLN diet-treated mice, did not display the uniform improvements in plasma lipids that were noted with a single injection, despite marked improvements in appetite, body weight, and plasma glucose levels (**Figure 4—figure supplement 3g–x**). Thus, while the positive effect of chronic asprosin neutralization on appetite, body weight and glucose homeostasis are clear; its positive

effect on plasma lipids appears inconsistent. This inconsistency may reflect adaptations and physiological differences in the three mouse models studied.

## Neutralization of asprosin using mAbs from distinct species and against distinct epitopes is equally protective

We wondered whether the positive metabolic effects of asprosin neutralization depended on a specific asprosin epitope or whether multiple epitopes could be targeted. We generated a new rabbit mAb against full-length, glycosylated human asprosin and a new fully human mAb derived from a human phage display library. Similar to previous results with the mouse mAb (*Duerrschmid et al., 2017*), both new mAbs (rabbit and fully human) produced a decrease in free asprosin levels that was directly proportional to the dose of the mAb used (*Figure 6—figure supplement 1*), suggesting potent neutralization ability. A single injection of each of the three mAbs resulted in significantly improved glucose tolerance (*Figure 6a*), and a reduction in food intake (*Figure 6b*) and body weight (*Figure 6c*) in DIO mice. Of these three mAbs, the mouse and human mAbs compete for binding to asprosin and so recognize the same or an overlapping epitope, whereas the rabbit mAb recognizes a distinct epitope, displaying no binding competition (*Figure 6d*). This indicates that, insofar as these two epitopes are concerned, the positive metabolic effects of asprosin neutralization are epitope-agnostic. We speculate that formation of an asprosin-mAb complex either accelerates asprosin disposal or prevents ligand–receptor interaction, effectively inhibiting the activity of endogenous asprosin. Given the multiple mAbs (against two distinct epitopes) employed, it is highly unlikely that off-target effects could account for the positive effects on metabolic health observed with all three mAbs.

## Pharmacokinetic parameters of asprosin neutralization

Using an antigen-capture ELISA, we measured the plasma concentrations of free mAbs at periodic intervals after injection to calculate their half-life in both endogenously high-asprosin (DIO mice) and low-asprosin ($Fbn1^{NPS/+}$ mice) conditions (*Figure 6e–g*). No mAbs against asprosin were detected before injection, whereas a calculated peak concentration of 0.1 mg/ml antigen-free mAb was observed 2 hr after injection. In DIO mice, the free mouse anti-asprosin mAb had the longest half-life, of approximately 11 days, the free rabbit anti-asprosin mAb had a half-life of approximately 5.6 days, and the free human anti-asprosin mAb had a half-life of approximately 3.2 days. Of note, the half-lives of these three mAbs were fairly similar in $Fbn1^{NPS/+}$ mice, indicating that the concentration of circulating asprosin does not appear to directly influence the rate of mAb clearance.

Asprosin-mAb affinity was measured on the Octet RED96 system (*Figure 6h–j*). The affinity ($K_D$) of the rabbit mAb to recombinant human asprosin was calculated as <10 pM from a $k_{on}$ of $1.11 \times 10^5$ $M^{-1}s^{-1}$ and a $k_{off} < 1 \times 10^{-7}$ $s^{-1}$ ($r^2 = 0.99$, $\chi^2 = 99.88$). The mouse mAb displayed a $K_D$ of $0.48 \pm 0.02$ nM with $k_{on}$ of $(1.45 \pm 0.003) \times 10^5$ $M^{-1}s^{-1}$ and $k_{off}$ of $(6.67 \pm 0.23) \times 10^{-5}$ $s^{-1}$ ($r^2 = 0.99$, $\chi^2 = 15.05$). The human mAb displayed a $K_D$ of $14.4 \pm 0.24$ nM with $k_{on}$ of $(6.76 \pm 0.07) \times 10^4$ $M^{-1}s^{-1}$ and $k_{off}$ of $(9.71 \pm 0.13) \times 10^{-4}$ $s^{-1}$ ($r^2 = 0.94$, $\chi^2 = 408.8$). Thus, the rabbit mAb has the highest affinity for asprosin and the human mAb the lowest, over a 1000-fold range. Nevertheless, all three mAbs were able to functionally neutralize asprosin in vivo.

## Anti-asprosin mAbs neutralize asprosin-mediated AgRP+neuron activation and hepatic cAMP signaling

To further our understanding of the neutralization mechanism of anti-asprosin mAbs, we measured AgRP+neuronal activity in response to recombinant asprosin and that preincubated with anti-asprosin mAbs. Similar to previously published effects of the mouse anti-asprosin mAb on AgRP+ neurons (*Duerrschmid et al., 2017*), the rabbit and fully human anti-asprosin mAbs reversibly and completely inhibited recombinant asprosin's activating effect on AgRP+neuron firing frequency and membrane potential (*Figure 7a–f*). Furthermore, a single injection of each of the three mAbs resulted in significant reduction in hepatic cAMP levels in fasting DIO mice (*Figure 7g,h*). These results indicate potent neutralizing activity of all three anti-asprosin mAbs against exogenous and endogenous asprosin, and at the level of both its functions (orexigenic and glucogenic).

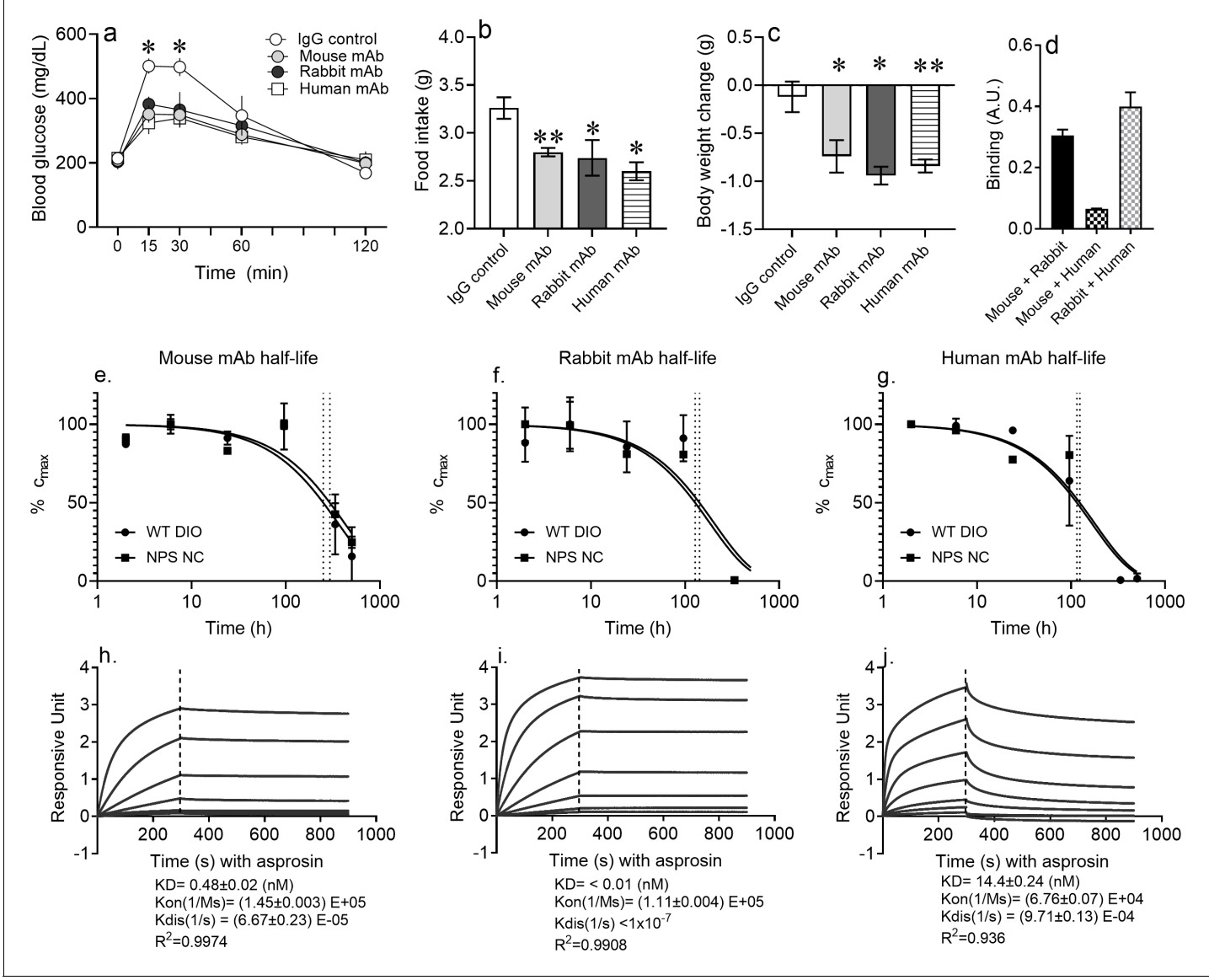

**Figure 6.** Pharmacokinetics of epitope-agnostic anti-asprosin mAbs from different sources. (a–c) Neutralization of asprosin using mAbs from different sources is equally protective. mAb generated using mouse immunization with a 28-mer asprosin peptide (mouse mAb), immunization of rabbits with recombinant full-length human asprosin (rabbit mAb), and recombinant mAb generated by panning phages from a naïve human antibody library (human mAb) were injected IP in 16-week-old male DIO mice (250 μg/mouse, ~5 mg/kg) and the indicated end points measured (n = 5/group). (d) *Epitope competition assay*: In a sandwich ELISA, asprosin captured by each mAb (mouse, rabbit, or human mAb) was detected by each of the three mAbs in a 3 × 3 matrix to determine competition for their respective epitopes. (e–g) Half-life of asprosin-neutralizing mAbs. Mouse models of 'high asprosin' (16-week-old male mice with diet-induced obesity; DIO), and 'low asprosin' (10-week-old male NPS mice) were injected with mouse, rabbit, or human mAb against asprosin 250 μg mAb in 500 μl 0.9% saline; n = 2/group. mAb levels were determined in mouse plasma collected at 2, 6, 24, 96, 336, and 504 hr post-injection to determine in vivo half-life of mAbs. (h–j) Anti-asprosin mAb binding affinity. Equilibrium dissociation constant (KD) of recombinant asprosin binding to anti-asprosin mAb was determined by a 1:1 binding model and use of global fitting method on Pall ForteBio's Octet RED96 system. Asterisk (*) indicate the range of alpha for an effect of mAb treatment as determined by t-test (two groups, one time point) or analysis of variance (ANOVA, sets involving multiple groups and time points; *Figure 6—source data 1*). *p<0.05, **p<0.01, ***p<0.001, and ****p<0.0001. The online version of this article includes the following source data and figure supplement(s) for figure 6:

**Source data 1.** Raw data and statistical analysis values for figure 6 and figure-supplements.
**Figure supplement 1.** Characterization of the asprosin-neutralizing antibodies.

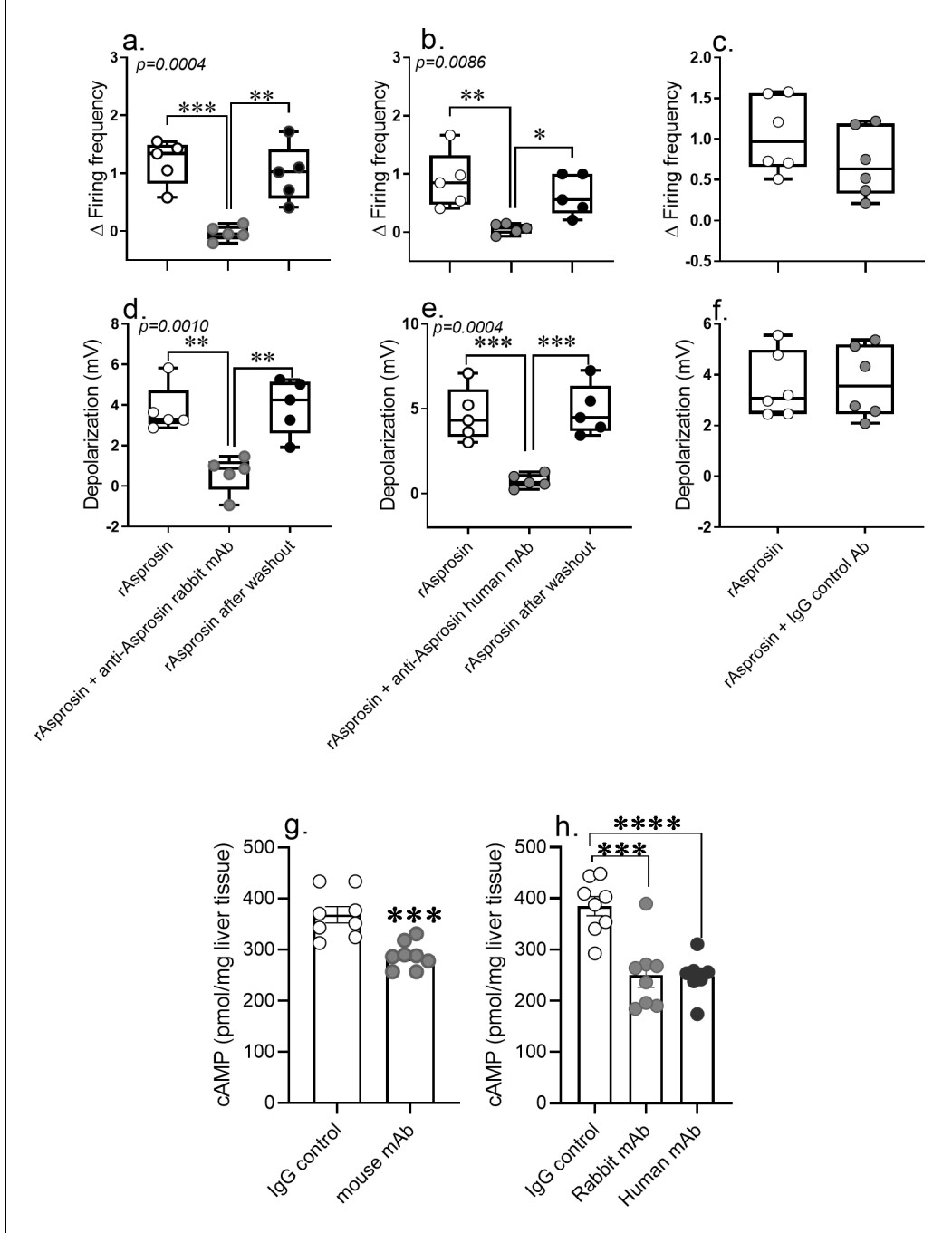

**Figure 7.** Anti-asprosin mAbs from different sources neutralize asprosin's orexigenic and glucogenic function by blunting AgRP+ neuronal firing and hepatic cAMP signaling. (**a–f**) *The anti-asprosin neutralizing antibodies reversibly inhibit asprosin's effect on AgRP+ neuronal activity.* Firing frequency (**a, b**) and membrane potential (**d, e**) of AgRP+ neurons were recorded in response to bacterial recombinant asprosin, asprosin preincubated with anti-asprosin rabbit and human mAb, and asprosin after washout. (**c, f**) Firing frequency response and membrane potential response of AgRP+ neurons were recorded in response to bacterial recombinant asprosin and IgG control antibody. Asterisk (*) indicate the range of alpha as determined by Sidak post hoc test followed by one-way analysis of variance (ANOVA; *Figure 7—source data 1*). *p<0.05, **p<0.01, ***p<0.001, and ****p<0.0001. (**g, h**) The anti-asprosin neutralizing antibodies blunt hepatic cAMP signaling. Sixteen-week-old male mice with diet-induced obesity (DIO) were injected with mouse, rabbit, or human mAb against asprosin (250 μg mAb in 500 μl 0.9% saline; n = 8/group). Three hour post-injection, hepatic cAMP levels were measured. Asterisk (*) indicate
*Figure 7 continued on next page*

*Figure 7 continued*

the range of alpha as determined by the two-tailed Student's t-test (*Figure 7—source data 1*). *p<0.05, **p<0.01, ***p<0.001, and ****p<0.0001.

The online version of this article includes the following source data for figure 7:

**Source data 1.** Raw data and statistical analysis values for *Figure 7*.

## Discussion

We recently discovered a novel glucogenic and orexigenic hormone, named asprosin, whose circulating levels are elevated in mice, rats, and humans with MS (*Romere et al., 2016*; *Li et al., 2019*; *Duerrschmid et al., 2017*; *Wang et al., 2020a*; *Ugur and Aydin, 2019*; *Alan et al., 2019*; *Zhang et al., 2019*; *Baykus et al., 2019*).

In this study, we have tested asprosin's physiological effects using three distinct gain-of-function (GOF) tools (adenoviral and adeno-associated viral vectors) and three independent loss-of-function (LOF) tools (three monoclonal antibodies targeting two distinct asprosin epitopes and derived from three distinct species). All three GOF tools produce an increase in blood glucose, appetite, and body weight in vivo. All three LOF tools produce the opposite in vivo. The fact that these six new experimental strategies produce consistent changes in blood glucose, appetite, and body weight in both directions (loss- or gain-of-function) leaves little doubt as to asprosin's function in physiology. In other words, these new results make it exceedingly difficult for asprosin's physiological effects to be explained by potential toxicity or reagent impurity. These diverse GOF and LOF tools also emphasize the high reproducibility of asprosin biology given that the proof obtained from such varied approaches bypasses technical difficulties inherent to any specific reagent/method such as recombinant proteins. To underscore this point, four outside groups have by now replicated asprosin's physiological effects in vivo (*Li et al., 2019*; *Yu et al., 2020*; *Hekim, 2021*; *Zhang et al., 2021*), and also expanded asprosin biology beyond our discoveries (modulation of muscle AMPK, identification of the liver asprosin receptor OR4M1, identification of a new asprosin-like hormone derived from FBN2, placensin). This is particularly important given the inability of one group (von Herrath et al.) to generate active recombinant asprosin (*von Herrath et al., 2019*). Our own experience with the unreliability of recombinant asprosin, whether made from mammalian or bacterial cells, makes it reasonable to assume that von Herrath et al. faced the same difficulties. There is no way to discern the quality of recombinant asprosin beyond measuring its effect in animals and cells. Its structure is not known, nor whether it needs a chaperone to attain correct stability/activity. The group that discovered the liver asprosin receptor (*Li et al., 2019*; *Li et al., 2019*) and reproduced the glucogenic effects of asprosin, remark on struggling with this issue – "Of note, the quality of Asprosin protein is critical for its activity. Similar to the results from *von Herrath et al., 2019*, we cannot purify the active His-Asprosin from inclusion bodies in *E. coli*. Therefore, we used GST-Asprosin in this study since GST-Asprosin is much more soluble." Our experience is similar to this, suggesting the need for a chaperone for attainment of reliable activity. As in this study, in the past we utilized several experimental strategies such as human/mouse genetic-mediated LOF, viral transduction-mediated GOF, and monoclonal antibody-mediated LOF to corroborate recombinant asprosin-mediated results, thereby thwarting the latter approach's unreliability. von Herrath et al. also attempted an alternative approach, reporting negative results with hydrodynamic gene delivery of asprosin. Unfortunately, neither this data, nor pertinent experimental details, such as use of a signal peptide, or most importantly, whether this strategy resulted in increased plasma asprosin levels, were shown. It is therefore impossible to ascertain the validity of this result. In summary, the problem with the von Herrath study is likely that the authors failed to view recombinant asprosin derived negative results skeptically given the inherent unreliability of that approach. Overall, the results presented here combined with the successful replication efforts by several independent groups render moot any remaining concerns about the physiological relevance of asprosin.

Asprosin neutralization resulted in reduced food intake and a parallel reduction in body weight and adipose mass in mice on a high-fat diet (DIO mice). A similar hypophagic and body weight reduction phenotype was also seen in more severe preclinical models of MS, such as mice on the AMLN diet and mice with a genetic loss of leptin signaling (*Lepr*$^{db/db}$). These results indicate that asprosin neutralization is effective independent of leptin and opens therapeutic avenues for a wide

range of cases of hyperphagia and obesity. Furthermore, the direction of improvement in response to asprosin neutralization was dependent on the stage and phase of MS. For example, chronic asprosin neutralization prevented weight gain or led to a significant weight loss, depending on the timing of treatment. Younger $Lepr^{db/db}$ mice, while still in the linear phase of body weight increase (12 weeks of age) (*Wang et al., 2020b*), were protected from weight gain upon asprosin neutralization. On the other hand, older $Lepr^{db/db}$ mice that have reached the plateaued weight phase (30 weeks of age) (*Wang et al., 2020b*) lost weight in response to asprosin neutralization.

Concurrent with decrease in appetite and body weight, anti-asprosin mAb therapy improved the MS-associated hyperglycemia in all preclinical models studied. Importantly, upon mAb treatment, the reduction in blood glucose in fasting DIO mice was not associated with changes in urine glucose levels, ruling out glucosuria as a potential contributing mechanism. Additionally, mAb-mediated improvements in hyperglycemia were independent of change in food intake given that mice were without access to food. Furthermore, a low $IC_{50}$ of 30–55 µg/mouse (~1.5 mg/kg) was determined for the four key features of MS (hyperphagia, obesity, hyperglycemia, and hypertriglyceridemia). The low $IC_{50}$, well within the spectrum of acceptable therapeutic mAb dosing, highlights the pharmacological inhibitory potency of asprosin neutralization for MS in its entirety, rather than improving only a particular manifestation of it (*Strik et al., 2018*). While the effect of escalating dose on appetite reduction is linear, it exhibits a defined upper and lower threshold when it comes to improvement of other attributes of MS such as blood glucose, insulin, triglycerides, and body weight. This may indicate arrival at a physiological equilibrium state and the impact of compensatory mechanisms to prevent further, potentially disastrous changes.

Interestingly, mAb treatment did not result in any changes in lean euglycemic WT mice except for a short-lived reduction in blood glucose (without crossing over into overt hypoglycemia). These results suggest the dependence of anti-asprosin mAb therapeutics on high endogenous asprosin levels and portends a strong safety profile. A strong safety profile and high specificity of anti-asprosin mAbs is reiterated by the complete lack of a response to mAb treatment in $Fbn1^{NPS/+}$ mice that display little circulating asprosin. It is also reiterated by the lack of confounding toxicity or side effects of asprosin neutralization (as measured by general activity, blood markers of inflammation, kidney, and liver health).

A single dose of the anti-asprosin mAb-improved MS-associated dyslipidemia in DIO mice as evidenced through a reduction in total cholesterol, LDL, triglycerides, and glycerol. This could potentially be explained by the glucose and insulin lowering effects of asprosin neutralization leading to suppression of lipogenesis (*Santoleri and Titchenell, 2019*; *Kersten, 2001*). With chronic neutralization, reductions in total cholesterol were noted in DIO and $Lepr^{db/db}$ mice. However, the effect on other lipid species was not as uniform as with single dosing, suggesting compensatory adaptations with chronic asprosin loss-of-function.

In addition to the mouse mAb, we generated rabbit and fully human mAbs for this study, both of which neutralized recombinant asprosin, similar to what was previously shown for the mouse mAb (*Duerrschmid et al., 2017*). The human mAb shares an epitope with the mouse mAb, while the rabbit mAb recognizes a distinct asprosin epitope. All three mAbs at a dose of 250 µg/mouse (~5 mg/kg) improved glucose tolerance and resulted in reduced food intake and body weight. This suggests that steric inhibition of asprosin's interaction with its receptor is either epitope-agnostic, or that the beneficial effects of asprosin neutralization depend on rapid asprosin disposal, or some combination of these two possibilities. Furthermore, all three mAbs showed an equilibrium dissociation constant ($K_D$) in the picomolar to low nanomolar range indicating high affinity mAb-asprosin binding (*Landry et al., 2015*).

Mechanistically, circulating asprosin elicits the glucogenic and orexigenic function by activating the hepatic G-protein-coupled receptor-associated cAMP-PKA signaling, and by activating the hypothalamic AgRP+neurons, respectively (*Romere et al., 2016*; *Li et al., 2019*; *Duerrschmid et al., 2017*). Anti-asprosin mAbs neutralize not only the circulating asprosin in different MS mouse models (*Romere et al., 2016*; *Duerrschmid et al., 2017*), but also asprosin-mediated hepatic cAMP signaling and AgRP neuron activity, demonstrating neutralization of asprosin action by all three mAbs at the level of both its functions – orexigenic and glucogenic. However, despite the high specificity, affinity, and long half-life displayed by the mAbs, maintenance of pharmacological effects required daily administration. A short 'effect-life', out of proportion to mAb half-life, has previously been reported for other mAbs against various circulating antigens (*Liu, 2014*;

*Rajan et al., 2017*; *Ovacik and Lin, 2018*). This short effect-life might be explained by high aspro-sin–mAb complex stability under various physiological conditions, leaving newly produced asprosin uninhibited (*Ovacik and Lin, 2018*; *Ryman and Meibohm, 2017*), thereby necessitating new mAb administration for continued pharmacological effect. In other words, there is sufficient precedence for this issue when it comes to circulating antigens and overcoming this requires a variety of mAb engineering approaches, which are under consideration for anti-asprosin mAbs at this time.

Conceptually however, the studies presented here offer promise for targeting asprosin in the treatment of MS, with planned lead optimization (to improve effect-life and humanize the lead candidate) (*Gui et al., 2019*), exploratory toxicity and efficacy studies in other species such as nonhuman primates, potentially leading to human trials. In addition, although the receptor for asprosin in the liver has been discovered recently (*Li et al., 2019*), the identity of its receptor in the CNS remains unknown. These receptor discovery efforts are essential for the development of orally bioavailable, small-molecule inhibitors of the asprosin pathway, which are validated by the ligand-targeting mAb approach presented here.

In summary, we provide evidence that demonstrates the potential of asprosin neutralization using multiple mAbs in vivo to treat MS. As opposed to current and past therapies that target satiety, this approach directly inhibits appetite and separately reduces the blood glucose burden by inhibiting hepatic glucose release. Thus, it is a dual-effect therapy that targets the two pillars of MS, over-nutri-tion and increased glucose burden. The results presented here, pre-clinically validating the pharma-cological inhibition of asprosin for the treatment of MS, are a source of high optimism moving forward.

## Acknowledgements

We thank Georgina Salazar, Andrew Pieper, Richard Premont, Mukesh Jain and Jonathan Stamler for critical reading of the manuscript. This work was supported by the Cancer Prevention and Research Institute of Texas (RP150551 and RP190561), the Welch Foundation (AU-0042–20030616 and I-1834), the NIDDK (DK102529, DK118290) and the Harrington Discovery Institute.

## Additional information

### Competing interests

Atul R Chopra: is a co-founder, director and officer of Vizigen, Inc, and Aceragen, Inc, and holds equity in both companies. The other authors declare that no competing interests exist.

### Funding

| Funder | Grant reference number | Author |
| --- | --- | --- |
| Cancer Prevention and Research Institute of Texas | RP150551 and RP190561 | Zhiqiang An |
| Welch Foundation | AU-0042-20030616 and I-1834 | Zhiqiang An |
| National Institute of Diabetes and Digestive and Kidney Diseases | DK102529 | Atul R Chopra |
| National Institute of Diabetes and Digestive and Kidney Diseases | DK118290 | Atul R Chopra |
| Harrington Discovery Institute | | Atul R Chopra |

The funders had no role in study design, data collection and interpretation, or the decision to submit the work for publication.

## Author contributions
Ila Mishra, Formal analysis, Investigation, Methodology, Writing - original draft; Clemens Duerrschmid, Investigation, Methodology, Writing - original draft; Zhiqiang Ku, Yang He, Wei Xin, Investigation, Methodology; Wei Xie, Elizabeth Sabath Silva, Jennifer Hoffman, Investigation; Ningyan Zhang, Supervision; Yong Xu, Resources, Supervision; Zhiqiang An, Resources, Supervision, Validation; Atul R Chopra, Conceptualization, Resources, Supervision, Funding acquisition, Writing - review and editing

## Author ORCIDs
Ila Mishra ⓘ https://orcid.org/0000-0002-9786-6358
Wei Xin ⓘ http://orcid.org/0000-0003-0987-0443
Zhiqiang An ⓘ http://orcid.org/0000-0001-9309-2335
Atul R Chopra ⓘ https://orcid.org/0000-0002-1304-3777

## Ethics
Animal experimentation: This study was performed in strict accordance with the recommendations in the Guide for the Care and Use of Laboratory Animals of the National Institutes of Health. All of the animals were handled according to approved institutional animal care and use committee (IACUC) protocols (#2018-0042) of the Case Western Reserve University. The protocol was approved by the Committee on the Ethics of Animal Experiments of Case Western Reserve University.

## Decision letter and Author response
Decision letter https://doi.org/10.7554/eLife.63784.sa1
Author response https://doi.org/10.7554/eLife.63784.sa2

# Additional files

## Supplementary files
• Transparent reporting form

## Data availability
All data analyzed during this study are included in the manuscript and supporting files.

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
