## [Decision Letter]

**Acceptance summary:**

The authors have previously identified asprosin as a novel orexigenic hormone, the present study provides proof of concept data on how targeting this hormone can result in potential benefits in conditions such as metabolic syndrome.

**Decision letter after peer review:**

Thank you for submitting your article "Asprosin Neutralizing Antibodies as a Treatment for Metabolic Syndrome" for consideration by *eLife*. Your article has been reviewed by 3 peer reviewers, including Carlos Isales as the Reviewing Editor and Reviewer #1, and the evaluation has been overseen by Mone Zaidi as the Senior Editor.

The reviewers have discussed the reviews with one another and the Reviewing Editor has drafted this decision to help you prepare a revised submission.

Summary:

Mishra et al. present data characterizing the effect of asprosin neutralizing antibodies on the parameters of metabolic syndrome (weight, glucose, lipids, etc). This group were the initial discoverers and characterized asprosin as a hormone that increases blood sugar and stimulates appetite. In their Nature Medicine 2017 article they also present data on a neutralizing antibody. In this follow up manuscript the group characterizes the impact of neutralizing monoclonal antibodies on metabolic parameters of three mouse models of obesity (DIO, NASH diet and Leptin receptor knockout). The translational focus of the manuscript is potential use of monoclonal antibodies against asprosin as a treatment of metabolic syndrome

Essential revisions:

A concern is the recent contradictory report by von Herrath et al. Cell Metabolism 2019 in which they failed to see the Asprosin's effect on food intake and glucose levels in mice. In the present manuscript the authors briefly discuss this by saying that this contradiction is "due to use of poor quality recombinant asprosin". But the contradictory paper in Cell Metabolism is from Novo Nordisk. It is important that the authors resolve this issue by providing a rigorous assessment of data reproducibility.

(1) One of the central issues is the specificity of the antibody action. The authors should demonstrate if the effect of the asprosin antibodies is blunted in mice that lack either asprosin or its receptor OR4M1.

(2) Additional characterization of the asprosin neutralizing effect of the AB is required.. It will be helpful to show the endogenous free asprosin levels at different time points after a single or repeated mAb injection. This result is also important to tell whether this mAb will cause other immune responses and side effects that might confound interpretation of the results.

(3) Previous studies from the authors' group show that asprosin acts on hepatocytes and triggers cAMP signaling. The authors should examine if the neutral antibodies blunt the cAMP signaling in DIO mice.

(4) Similarly, asprosin was shown to stimulate AgRP+ neurons. The authors need to demonstrate the effect of asprosin antibodies on AgRP+ neuronal activity.

(5) in Figure 1 the blood glucose drops independent of food intake is this all related to decreased hepatic glucose output or are there any effect on urine. Was urinary glucose measured? Is there increased glycosuria?

(6) Most of the bodyweight data are presented as "body-weight change". However, the authors should present them as whole-body weight. In Figure 3 (a, e, j) and Figure 4 (a, e, I, m). please show body weight to rule out the stress or side effects cause by virus injection. For DIO mice, 14 days IgG injection also caused weight loss; for db/db mice, IgG injection increased body weight. Please discuss. In previous papers the authors discuss the increased lean body mass when asprosin is not present. There is no body composition data in this study. Was there any body composition differences with the antibody among the different mouse models (e.g DIO vs Nash diet)?

(7) Although adenovirus and AAV are widely used for in vivo protein overexpression, it is important to show here that endogenous asprosin levels were increased after virus injection and decreased after antibody neutralization.

(8) In Figure 5, more data on liver weight, histology, etc. is required to support their conclusion on liver health. The current data from three different mice models are very contradictive, this can be caused by the side effect or off-target effect of this mAb.

(9) In Figure 5 there are a number of inflammatory markers which can vary according to the model. What about anti-inflammatory markers (cortisol, IL-10 etc) would be helpful to get a better picture of physiologic changes.

(10) In Figure 6, it is important to demonstrate the neutralizing effect of the mAbs.

---

## [Author Response]

Essential revisions:A concern is the recent contradictory report by von Herrath et al. Cell Metabolism 2019 in which they failed to see the Asprosin's effect on food intake and glucose levels in mice. In the present manuscript the authors briefly discuss this by saying that this contradiction is "due to use of poor quality recombinant asprosin". But the contradictory paper in Cell Metabolism is from Novo Nordisk. It is important that the authors resolve this issue by providing a rigorous assessment of data reproducibility.

In this study, we have tested asprosin’s physiological effects using three distinct gain-of-function (GOF) tools (adenoviral and adeno-associated viral vectors) and three independent loss-of-function (LOF) tools (three monoclonal antibodies targeting two distinct asprosin epitopes and derived from three distinct species). All three GOF tools produce an increase in blood glucose, appetite, and body weight *in vivo*. All three LOF tools produce the opposite *in vivo*. The fact that these six independent experimental strategies produce consistent changes in blood glucose, appetite and body weight in both directions (loss- or gain-of-function) leaves little doubt as to asprosin’s function in physiology. In other words, these new results make it exceedingly difficult for asprosin’s physiological effects to be explained by potential toxicity or reagent impurity. These diverse GOF and LOF tools also emphasize the high reproducibility of asprosin biology given that the proof obtained from such varied approaches bypasses technical difficulties inherent to any specific reagent/method such as recombinant proteins. To underscore this point, to our knowledge four independent groups have by now replicated asprosin’s physiological effects *in vivo* and also expanded asprosin biology beyond our discoveries (modulation of muscle AMPK, identification of the liver asprosin receptor OR4M1, identification of a new asprosin-like hormone derived from FBN2, placensin)^1-4^. This is particularly important given the inability of one group (von Herrath et al.) to generate active recombinant asprosin^5^. Our own experience with the unreliability of recombinant asprosin, whether made from mammalian or bacterial cells, makes it reasonable to assume that von Herrath et al. faced the same difficulties. There is no way to discern the quality of recombinant asprosin beyond measuring its effect in animals and cells. Its structure is not known, nor whether it needs a chaperone to attain correct stability/activity. The group that discovered the liver asprosin receptor (Li et al., 2019) and replicated the glucogenic effects of asprosin^1^, remark on struggling with this issue – “Of note, the quality of Asprosin protein is critical for its activity. Similar to the results from von Herrath et al. (2019), we cannot purify the active His-Asprosin from inclusion bodies in *E. coli*. Therefore, we used GST-Asprosin in this study since GST-Asprosin is much more soluble.” Our experience is similar to this, suggesting the need for a chaperone for attainment of reliable activity. As in this study, in the past we utilized several experimental strategies such as human/mouse genetic-mediated LOF, viral transduction-mediated GOF, and monoclonal antibody-mediated LOF to corroborate recombinant asprosin-mediated results, thereby thwarting the latter approach’s unreliability. von Herrath et al. also attempted an alternative approach, reporting negative results with hydrodynamic gene delivery of asprosin^5^. Unfortunately, neither this data, nor pertinent experimental details, such as use of a signal peptide, or most importantly, whether this strategy resulted in increased plasma asprosin levels, were shown. It is therefore impossible to ascertain the validity of this result. In summary, the problem with the von Herrath study is likely that the authors failed to view recombinant asprosin-derived negative results skeptically given the inherent unreliability of that approach. Overall, the results presented here combined with the successful replication efforts by several independent groups render moot any remaining concerns about the physiological relevance of asprosin.

Given the importance of this issue for the field, we have included it in the Discussion section of the manuscript (line 253-291).

(1) One of the central issues is the specificity of the antibody action. The authors should demonstrate if the effect of the asprosin antibodies is blunted in mice that lack either asprosin or its receptor OR4M1.

This is an astute suggestion. We now present data of anti-asprosin mAb treatment of *Fbn1^NPS/+^*mice which serve as a genetic model of asprosin deficiency. *Fbn1^NPS/+^* mice were totally unresponsive to the mAb treatment, with no effect at all on blood glucose, 24h cumulative food intake and body weight (Figure 1—figure supplement 4). This demonstrates the inability of asprosin-neutralization to produce a physiological effect in mice lacking circulating asprosin, in contrast to the marked effects seen in mouse models with elevated circulating asprosin. This clearly shows that the glucose-, appetite-, and weight-reducing effects of anti-asprosin mAbs occur due to neutralization of asprosin rather than due to potential non-specificity or toxicity.

(2) Additional characterization of the asprosin neutralizing effect of the AB is required.. It will be helpful to show the endogenous free asprosin levels at different time points after a single or repeated mAb injection. This result is also important to tell whether this mAb will cause other immune responses and side effects that might confound interpretation of the results.

We have previously shown that with increasing concentrations of the mouse anti-asprosin mAb used in this study, there is a decline in the concentration of free asprosin (please see Suppl. Figure 8C, Duerrschmid C et al., 2017)^6^. We present new data showing dose-dependent decrease in free asprosin using the other two mAbs – rabbit and fully human (Figure 6—figure supplement 1). Additionally, we have previously shown reduction in endogenous free circulating asprosin at multiple time points after exposure to the mouse anti-asprosin mAb used in this study, in DIO and *Ob/Ob* mice (please see figure 7c,g, Romere et al., 2016)^7^.

To further characterize the mechanism of action of the anti-asprosin mAbs at the level of the orexigenic and glucogenic functions of asprosin, we now show the potent neutralizing effect of all three anti-asprosin mAbs on asprosin-mediated AgRP neuronal activity and hepatic cAMP content (Figure 7, please also see response to comment #3 and 5). These results demonstrate that asprosin neutralization specifically inhibits the asprosin pathway at the level of both asprosin functions.

Furthermore, to ensure that weight loss, reduced hunger and glucose normalizing effects are the result of asprosin neutralization, and not potential toxicity or off-target effects, we measured different markers of sickness – proinflammatory cytokines (IL1β/TNFα), anti-inflammatory cytokines (TGFβ and IL10), markers of hepatic health (plasma ALT and AST levels), markers of renal health (Creatinine and BUN), and two measures of general activity: pedestrian locomotor activity and intentional wheel running activity. We did not observe any indication of potential toxicity/sickness using this multi-layered approach. Please see Figure 1—figure supplement 5.

(3) Previous studies from the authors' group show that asprosin acts on hepatocytes and triggers cAMP signaling. The authors should examine if the neutral antibodies blunt the cAMP signaling in DIO mice.

We now present new data showing effects of asprosin neutralization, using all three mAbs on hepatic cAMP levels (Figure 7). All three mAbs significantly reduce hepatic cAMP content *in vivo* suggesting successful neutralization of endogenous hepatic asprosin signaling.

(4) Similarly, asprosin was shown to stimulate AgRP+ neurons. The authors need to demonstrate the effect of asprosin antibodies on AgRP+ neuronal activity.

This is an important suggestion. While the neutralizing effect of the mouse mAb on AgRP neuron activity has previously been shown by us (please see Supplementary Figure 9, Duerrschmid C et al., 2017)^6^, we now present new data showing the neutralizing effect of the other two mAbs (rabbit and fully human) on AgRP neuronal activity (Figure 7). All three mAbs completely neutralize the ability of asprosin to activate AgRP neurons.

(5) in Figure 1 the blood glucose drops independent of food intake is this all related to decreased hepatic glucose output or are there any effect on urine. Was urinary glucose measured? Is there increased glycosuria?

This is an excellent suggestion, particularly given the anti-diabetic effects of SGLT2 inhibitors in the clinic. We measured but did not find any effect of asprosin neutralization on glucosuria in DIO mice (Figure 1—figure supplement 2), specifically at timepoints that coincide with reductions in blood glucose. Thus, we have ruled out enhancement in glucosuria as a potential mechanism for the anti-diabetic effects of asprosin neutralization.

(6) Most of the bodyweight data are presented as "body-weight change". However, the authors should present them as whole-body weight. In Figure 3 (a, e, j) and Figure 4 (a, e, I, m). please show body weight to rule out the stress or side effects cause by virus injection. For DIO mice, 14 days IgG injection also caused weight loss; for db/db mice, IgG injection increased body weight. Please discuss. In previous papers the authors discuss the increased lean body mass when asprosin is not present. There is no body composition data in this study. Was there any body composition differences with the antibody among the different mouse models (e.g DIO vs Nash diet)?

Body weight change measurement equalizes the average weight of the two groups (IgG versus anti-asprosin mAb) prior to treatment, enabling a clear graphical depiction of the change elicited by the two agents from the same starting point. However, in response to this suggestion, we now present whole-body weight in Figure 3—figure supplement 2, figure 4—figure supplement 1 and 2for all experiments of figures 3 and 4. The body weight trajectories in response to the mAbs show no sudden drops or other changes that would suggest stress or toxicity.

For DIO and AMLN-diet treated mice, where 14 days of anti-asprosin mAb treatment caused weight loss, the body weight had already reached a plateau prior to treatment. In contrast, for 12-week-old *Db/Db* mice, where 14 days of anti-asprosin mAb treatment prevented weight gain, mice were still gaining weight prior to treatment. In the latter experiment, it is not the IgG that is making the mice gain weight but rather their own intrinsic biology at 12 weeks of age. Asprosin neutralization simply prevents that weight gain. To clarify this point, we have repeated the *Db/Db* experiment, this time with 30-week-old mice where their body weight had plateaued prior to treatment (figure 4—figure supplement 3). In this setting, asprosin neutralization results in weight loss, much like the DIO experiment. The only experiment where IgG showed a minor effect on body weight is in 30-week-old DIO mice (Figure 4e) where IgG injected mice lost some weight at the beginning of the experiment. We attribute this to handling-related transient stress imposed on the mice as the body weight returned to normal midway through the experiment. However, we have repeated the DIO experiment in 16-week-old mice (on HFD for 12 weeks starting at 4 weeks of age), and again found a significant weight reduction produced by the anti-asprosin mAb, and no impact of IgG on body weight (figure 4—figure supplement 3).

Further, we present MRI body composition analysis of body weight, lean mass, subcutaneous and intraperitoneal fat mass in DIO mice before and after 14 days of asprosin neutralization (Figure 5). *Fbn1^NPS/+^* mice have asprosin deficiency in addition to Marfan syndrome (from mutant fibrillin-1), leading to a reduction in lean mass (a Marfan syndrome effect) in addition to a reduction in adipose mass^6^. In contrast, anti-asprosin mAb treatment in DIO mice results in a marked reduction in adipose mass without affecting lean mass at all. That is because asprosin neutralization does not affect fibrillin-1 while inhibiting circulating asprosin. That is a key difference between *Fbn1^NPS/+^* mice (which show effects of both asprosin and fibrillin-1 inhibition) and mice treated with anti-asprosin mAbs (which show effects of just asprosin inhibition).

(7) Although adenovirus and AAV are widely used for in vivo protein overexpression, it is important to show here that endogenous asprosin levels were increased after virus injection and decreased after antibody neutralization.

In figure 4—figure supplement 1e,1j,1o we show that *in vivo* circulating asprosin levels were significantly increased in mice transduced with asprosin expressing adeno- and adeno-associated viruses, each equipped with an IL2 signal peptide to promote secretion. Please see our response to comment#2 where we have described reductions in free circulating asprosin upon anti-asprosin mAb treatment.

(8) In Figure 5, more data on liver weight, histology, etc. is required to support their conclusion on liver health. The current data from three different mice models are very contradictive, this can be caused by the side effect or off-target effect of this mAb.

We fully agree with reviewer’s points #9 and 10, in that assessment of mRNA expression and liver triglycerides alone is insufficient to convincingly claim effects of asprosin neutralization on liver health and NASH. A thorough analysis of liver mass normalized to body mass, NAFLD activity scoring (NAS) based on hepatocyte steatosis, inflammation and ballooning, corresponding H&E and Masson’s trichrome stained liver sections for fibrosis, inflammation at mRNA and protein levels, leukocyte infiltration, and gross hepatic steatosis is warranted to conclude any effects of mAb treatment on liver health. Further, it needs to be ascertained whether chronic mAb treatment can prevent or potentially reverse NASH. If any effects are seen, the timeline, trajectory, and persistence of such effects needs to be ascertained. Additionally, a single model of NASH would appear to be insufficient to arrive at a convincing assessment of the efficacy of asprosin neutralization. We would need at least two models, preferably one genetic and one environmental.

At present, the above seems outside the scope of this study on metabolic syndrome. We expect that ascertaining the effects of asprosin neutralization on liver health and NASH will require at least as much data as we have presented here on metabolic syndrome in order to make a convincing case, suggesting an independent study. To this effect, we have decided to focus exclusively on metabolic syndrome in this report and build the evidence for NASH in a separate study based on the feedback received here.

(9) In Figure 5 there are a number of inflammatory markers which can vary according to the model. What about anti-inflammatory markers (cortisol, IL-10 etc) would be helpful to get a better picture of physiologic changes.

Please see the above response to comment#8.

(10) In Figure 6, it is important to demonstrate the neutralizing effect of the mAbs.

Please see response #4 and 5. We present new data showing the effects of asprosin neutralization, using all three mAbs, on hepatic cAMP levels (Figure 7). All three mAbs significantly reduce hepatic cAMP content *in vivo* suggesting successful neutralization of endogenous hepatic asprosin signaling^1,7^. While the neutralizing effect of the mouse mAb on AgRP neuron activity has previously been shown by us (please see Supplementary Figure 9, Duerrschmid C et al., 2017)^6^, we now present new data showing the neutralizing effect of the other two mAbs (rabbit and fully human) on AgRP neuronal activity (Figure 7). All three mAbs completely neutralize the ability of asprosin to activate AgRP neurons. These results demonstrate that asprosin neutralization specifically inhibits the asprosin pathway at the level of both asprosin functions (glucogenic and orexigenic).

References:

1. Li E, Shan H, Chen L, Long A, Zhang Y, Liu Y, Jia L, Wei F, Han J, Li T, Liu X, Deng H, Wang Y. OLFR734 mediates glucose metabolism as a receptor of asprosin. Cell Metab. 2019; 30(2):319-328.e8. doi: 10.1016/j.cmet.2019.05.022. PMID: 31230984.

2. Yu Y, He JH, Hu LL, Jiang LL, Fang L, Yao GD, Wang SJ, Yang Q, Guo Y, Liu L, Shang T, Sato Y, Kawamura K, Hsueh AJ, Sun YP. Placensin is a glucogenic hormone secreted by human placenta. EMBO Rep. 2020; 21(6):e49530. doi: 10.15252/embr.201949530. PMID: 32329225.

3. Hekim MG, Kelestemur MM, Bulmus FG, Bilgin B, Bulut F, Gokdere E, Ozdede MR, Kelestimur H, Canpolat S, Ozcan M. Asprosin, a novel glucogenic adipokine: a potential therapeutic implication in diabetes mellitus. Arch Physiol Biochem. 2021; 4:1-7. doi: 10.1080/13813455.2021.1894178. PMID: 33663304.

4. Zhang Y, Zhu Z, Zhai W, Bi Y, Yin Y, Zhang W. Expression and purification of asprosin in Pichia pastoris and investigation of its increase glucose uptake activity in skeletal muscle through activation of AMPK. Enzyme Microb Technol. 2021;144:109737. doi: 10.1016/j.enzmictec.2020.109737. PMID: 33541572.

5. von Herrath M, Pagni PP, Grove K, Christoffersson G, Tang-Christensen M, Karlsen AE, Petersen JS. Case reports of pre-clinical replication studies in metabolism and diabetes. Cell Metab. 2019; 29(4):795-802. doi: 10.1016/j.cmet.2019.02.004. PMID: 30879984.

6. Duerrschmid C, He Y, Wang C, Li C, Bournat JC, Romere C, Saha PK, Lee ME, Phillips KJ, Jain M, Jia P, Zhao Z, Farias M, Wu Q, Milewicz DM, Sutton VR, Moore DD, Butte NF, Krashes MJ, Xu Y, Chopra AR. Asprosin is a centrally acting orexigenic hormone. Nat Med. 2017; 23(12):1444-1453. doi: 10.1038/nm.4432. PMID: 29106398.

7. Romere C, Duerrschmid C, Bournat J, Constable P, Jain M, Xia F, Saha PK, Del Solar M, Zhu B, York B, Sarkar P, Rendon DA, Gaber MW, LeMaire SA, Coselli JS, Milewicz DM, Sutton VR, Butte NF, Moore DD, Chopra AR. Asprosin, a fasting-induced glucogenic protein hormone. Cell. 2016; 165(3):566-79. doi: 10.1016/j.cell.2016.02.063. PMID: 27087445.